# Beyond Exponential Graph: Communication-Efficient Topologies for Decentralized Learning via Finite-time Convergence

**Yuki Takezawa**[1,2]*, **Ryoma Sato**[1,2]*, **Han Bao**[1,2], **Kenta Niwa**[3], **Makoto Yamada**[2]
[1]Kyoto University, [2]OIST, [3]NTT Communication Science Laboratories

## Abstract

Decentralized learning has recently been attracting increasing attention for its applications in parallel computation and privacy preservation. Many recent studies stated that the underlying network topology with a faster consensus rate (a.k.a. spectral gap) leads to a better convergence rate and accuracy for decentralized learning. However, a topology with a fast consensus rate, e.g., the exponential graph, generally has a large maximum degree, which incurs significant communication costs. Thus, seeking topologies with both a fast consensus rate and small maximum degree is important. In this study, we propose a novel topology combining both a fast consensus rate and small maximum degree called the BASE-$(k + 1)$ GRAPH. Unlike the existing topologies, the BASE-$(k + 1)$ GRAPH enables all nodes to reach the exact consensus after a finite number of iterations for any number of nodes and maximum degree $k$. Thanks to this favorable property, the BASE-$(k + 1)$ GRAPH endows Decentralized SGD (DSGD) with both a faster convergence rate and more communication efficiency than the exponential graph. We conducted experiments with various topologies, demonstrating that the BASE-$(k + 1)$ GRAPH enables various decentralized learning methods to achieve higher accuracy with better communication efficiency than the existing topologies. Our code is available at `https://github.com/yukiTakezawa/BaseGraph`.

## 1 Introduction

Distributed learning, which allows training neural networks in parallel on multiple nodes, has become an important paradigm owing to the increased utilization of privacy preservation and large-scale machine learning. In a centralized fashion, such as All-Reduce and Federated Learning [8, 9, 16, 26, 27], all or some selected nodes update their parameters by using their local dataset and then compute the average parameter of these nodes, although computing the average of many nodes is the major bottleneck in the training time [18, 19, 23]. To reduce communication costs, decentralized learning gains significant attention [11, 18, 24]. Because decentralized learning allows nodes to exchange parameters only with a few neighbors in the underlying network topology, decentralized learning is more communication efficient than All-Reduce and Federated Learning.

While decentralized learning can improve communication efficiency, it may degrade the convergence rate and accuracy due to its sparse communication characteristics [11, 48]. Specifically, the smaller the maximum degree of an underlying network topology is, the fewer the communication cost becomes [33, 39]; meanwhile, the faster the consensus rate (a.k.a. spectral gap) of a topology is, the faster the convergence rate of decentralized learning becomes [11]. Thus, developing a topology with both a fast consensus rate and small maximum degree is essential for decentralized learning. Table 1 summarizes the properties of various topologies. For instance, the ring and exponential graph are

---

*Equal Contribution

Table 1: Comparison among different topologies with $n$ nodes. The definition of the consensus rate and finite-time convergence is shown in Sec. 3.

| Topology | Consensus Rate | Connection | Maximum Degree | #Nodes $n$ |
|---|---|---|---|---|
| Ring [28] | $1 - O(n^{-2})$ | Undirected | 2 | $\forall n \in \mathbb{N}$ |
| Torus [28] | $1 - O(n^{-1})$ | Undirected | 4 | $\forall n \in \mathbb{N}$ |
| Exp. [43] | $1 - O((\log_2(n))^{-1})$ | Directed | $\lceil \log_2(n) \rceil$ | $\forall n \in \mathbb{N}$ |
| 1-peer Exp. [43] | $O(\log_2(n))$-finite time conv. | Directed | 1 | A power of 2 |
| 1-peer Hypercube [31] | $O(\log_2(n))$-finite time conv. | Undirected | 1 | A power of 2 |
| **Base-$(k+1)$ Graph (ours)** | $O(\log_{k+1}(n))$-finite time conv. | Undirected | $k$ | $\forall n \in \mathbb{N}$ |

commonly used [1, 3, 12, 23]. The ring is a communication-efficient topology because its maximum degree is two but its consensus rate quickly deteriorates as the number of nodes $n$ increases [28]. The exponential graph has a fast consensus rate, which does not deteriorate much as $n$ increases, but it incurs significant communication costs because its maximum degree increases as $n$ increases [43]. Thus, these topologies sacrifice either communication efficiency or consensus rate.

Recently, the 1-peer exponential graph [43] and 1-peer hypercube graph [31] were proposed as topologies that combine both a small maximum degree and fast consensus rate (see Sec. C.4 for examples). As Fig. 1 shows, in the ring and exponential graph, node parameters only reach the consensus asymptotically by repeating exchanges of parameters with neighbors. Contrarily, in the 1-peer exponential and 1-peer hypercube graphs, parameters reach the exact consensus after a finite number of iterations when $n$ is a power of 2 (see Fig. 21 in Sec. F.2). Thanks to this property of finite-time convergence, the 1-peer exponential and 1-peer hypercube graphs enable Decentralized SGD (DSGD) [18] to converge at the same convergence rate as the exponential graph when $n$ is a power of 2, even though the maximum degree of the 1-peer exponential and 1-peer hypercube graphs is only one [43].

However, this favorable property only holds when $n$ is a power of 2. When $n$ is not a power of 2, the 1-peer hypercube graph cannot be constructed, and the 1-peer exponential graph only reaches the consensus asymptotically as well as the ring and exponential graph, as Fig. 1 illustrates. Thus, the 1-peer exponential and 1-peer hypercube graphs cannot enable DSGD to converge as fast as the exponential graph when $n$ is not a power of 2. Moreover, even if $n$ is a power of 2, the 1-peer hypercube and 1-peer exponential graphs still cannot enable DSGD to converge faster than the exponential graph.

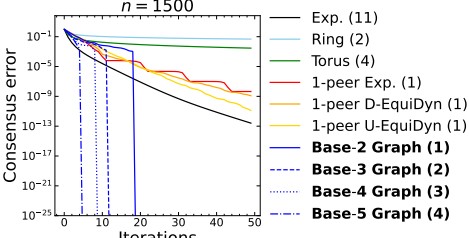

Figure 1: Comparison of consensus rate. See Sec. 6 for detailed experimental settings. The number in the bracket is the maximum degree.

In this study, we ask the following question: *Can we construct topologies that provide DSGD with both a faster convergence rate and better communication efficiency than the exponential graph for any number of nodes?* Our work provides the affirmative answer by proposing the BASE-$(k+1)$ GRAPH,[2] which is finite-time convergence for any number of nodes $n$ and maximum degree $k$ (see Fig. 1). Thanks to this favorable property, the BASE-2 GRAPH enables DSGD to converge faster than the ring and torus and as fast as the exponential graph for any $n$, while the BASE-2 GRAPH is more communication-efficient than the ring, torus, and exponential graph because its maximum degree is only one. Furthermore, when $2 \le k < \lceil \log_2(n) \rceil$, the BASE-$(k+1)$ GRAPH enables DSGD to converge faster with fewer communication costs than the exponential graph because the maximum degree of the BASE-$(k+1)$ GRAPH is still less than that of the exponential graph. Experimentally, we compared the BASE-$(k+1)$ GRAPH with various existing topologies, demonstrating that the BASE-$(k+1)$ GRAPH enables various decentralized learning methods to more successfully reconcile accuracy and communication efficiency than the existing topologies.

## 2 Related Work

**Decentralized Learning.** The most widely used decentralized learning methods are DSGD [18] and its adaptations [1, 2, 19]. Many researchers have improved DSGD and proposed DSGD with momentum [4, 20, 44, 45], communication compression methods [6, 10, 23, 35, 38], etc. While DSGD is a simple and efficient method, DSGD is sensitive to data heterogeneity [36]. To mitigate this issue, various methods have been proposed, which can eliminate the effect of data heterogeneity on the convergence rate, including gradient tracking [21, 29, 30, 34, 42, 47], $D^2$ [36], etc. [17, 37, 46].

---

[2]Note that the maximum degree of the BASE-$(k+1)$ GRAPH is not $k+1$, but at most $k$.

**Effect of Topologies.** Many prior studies indicated that topologies with a fast consensus rate improve the accuracy of decentralized learning [11, 13, 25, 39]. For instance, DSGD and gradient tracking converge faster as the topology has a faster consensus rate [18, 34]. Zhu et al. [48] revealed that topology with a fast consensus rate improves the generalization bound of DSGD. Especially when the data distributions are statistically heterogeneous, the topology with a fast consensus rate prevents the parameters of each node from drifting away and can improve accuracy [11, 34]. However, communication costs increase as the maximum degree increases [33, 43]. Thus, developing a topology with a fast consensus rate and small maximum degree is important for decentralized learning.

## 3 Preliminary and Notation

**Notation.** A graph $G$ is represented by $(V, E)$ where $V$ is a set of nodes and $E$ is a set of edges. If $G$ is a graph, $V(G)$ (resp. $E(G)$) denotes the set of nodes (resp. edges) of $G$. For any $a_1, \cdots, a_n$, $(a_1, \cdots, a_n)$ denotes the ordered set. An empty (ordered) set is denoted by $\emptyset$. For any $n \in \mathbb{N}$, let $[n] := \{1, \cdots, n\}$. For any $n, a \in \mathbb{N}$, $\mathrm{mod}(a, n)$ is the remainder of dividing $a$ by $n$. $\|\cdot\|_F$ denotes Frobenius norm, and $\mathbf{1}_n$ denotes an $n$-dimensional vector with all ones.

**Topology.** Let $G$ be an underlying network topology with $n$ nodes, and $\boldsymbol{W} \in [0, 1]^{n \times n}$ be a mixing matrix associated with $G$. That is, $W_{ij}$ is the weight of the edge $(i, j)$, and $W_{ij} > 0$ if and only if $(i, j) \in E(G)$. Most of the decentralized learning methods require $\boldsymbol{W}$ to be doubly stochastic (i.e., $\boldsymbol{W}\mathbf{1}_n = \mathbf{1}_n$ and $\boldsymbol{W}^\top \mathbf{1}_n = \mathbf{1}_n$) [18, 20, 29, 36]. Then, the consensus rate of $G$ is defined below.

**Definition 1.** *Let $\boldsymbol{W}$ be a mixing matrix associated with a graph $G$ with $n$ nodes. Let $\boldsymbol{x}_i \in \mathbb{R}^d$ be a parameter that node $i$ has. Let $\boldsymbol{X} := (\boldsymbol{x}_1, \cdots, \boldsymbol{x}_n) \in \mathbb{R}^{d \times n}$ and $\bar{\boldsymbol{X}} := \frac{1}{n}\boldsymbol{X}\mathbf{1}_n\mathbf{1}_n^\top$. The consensus rate $\beta \in [0, 1)$ is the smallest value that satisfies $\left\|\boldsymbol{X}\boldsymbol{W} - \bar{\boldsymbol{X}}\right\|_F^2 \leq \beta^2 \left\|\boldsymbol{X} - \bar{\boldsymbol{X}}\right\|_F^2$ for any $\boldsymbol{X}$.*

Thanks to $\beta \in [0, 1)$, $\boldsymbol{x}_i$ asymptotically converge to consensus $\frac{1}{n}\sum_{i=1}^n \boldsymbol{x}_i$ by repeating parameter exchanges with neighbors. However, this does not mean that all nodes reach the exact consensus within a finite number of iterations except when $\beta = 0$, that is, when $G$ is fully connected. Then, utilizing time-varying topologies, Ying et al. [43] and Shi et al. [31] aimed to obtain sequences of graphs that can make all nodes reach the exact consensus within finite iterations and proposed the 1-peer exponential and 1-peer hypercube graphs respectively (see Sec. C.4 for illustrations).

**Definition 2.** *Let $(G^{(1)}, \cdots, G^{(m)})$ be a sequence of graphs with the same set of nodes (i.e., $V(G^{(1)}) = \cdots = V(G^{(m)})$). Let $n$ be the number of nodes. Let $\boldsymbol{W}^{(1)}, \cdots, \boldsymbol{W}^{(m)}$ be mixing matrices associated with $G^{(1)}, \cdots, G^{(m)}$, respectively. Suppose that $\boldsymbol{W}^{(1)}, \cdots, \boldsymbol{W}^{(m)}$ satisfy $\boldsymbol{X}\boldsymbol{W}^{(1)}\boldsymbol{W}^{(2)} \cdots \boldsymbol{W}^{(m)} = \bar{\boldsymbol{X}}$ for any $\boldsymbol{X} \in \mathbb{R}^{d \times n}$, where $\bar{\boldsymbol{X}} = \frac{1}{n}\boldsymbol{X}\mathbf{1}_n\mathbf{1}_n^\top$. Then, $(G^{(1)}, \cdots, G^{(m)})$ is called $m$-finite time convergence or an $m$-finite time convergent sequence of graphs.*

Because Definition 2 assumes that $V(G^{(1)}) = \cdots = V(G^{(m)})$ holds, we often write a sequence of graphs $(G^{(1)}, \cdots, G^{(m)})$ as $(E(G^{(1)}), \cdots, E(G^{(m)}))$ using a slight abuse of notation. Additionally, in the following section, we often abbreviate the weights of self-loops because they are uniquely determined due to the condition that the mixing matrix is doubly stochastic.

## 4 Construction of Finite-time Convergent Sequence of Graphs

In this section, we propose the BASE-$(k + 1)$ GRAPH, which is finite-time convergence for any number of nodes $n \in \mathbb{N}$ and maximum degree $k \in [n - 1]$. Specifically, we consider the setting where node $i$ has a parameter $\boldsymbol{x}_i$ and propose a graph sequence whose maximum degree is at most $k$ that makes all nodes reach the exact consensus $\frac{1}{n}\sum_{i=1}^n \boldsymbol{x}_i$. To this end, we first propose the $k$-PEER HYPER-HYPERCUBE GRAPH, which is finite-time convergence when $n$ does not have prime factors larger than $k + 1$. Using it, we propose the SIMPLE BASE-$(k+1)$ GRAPH and BASE-$(k+1)$ GRAPH, which are finite-time convergence for any $n$.

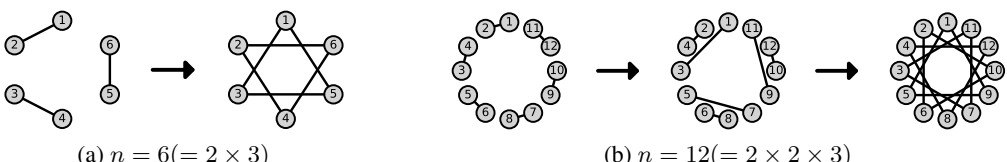

(a) $n = 6 (= 2 \times 3)$        (b) $n = 12 (= 2 \times 2 \times 3)$

Figure 2: Illustration of the 2-PEER HYPER-HYPERCUBE GRAPH.

---

**Algorithm 1** $k$-PEER HYPER-HYPERCUBE GRAPH $\mathcal{H}_k(V)$

---

1: **Input:** the set of nodes $V \coloneqq \{v_1, \cdots, v_n\}$ and number of nodes $n$.
2: Decompose $n$ as $n = n_1 \times \cdots \times n_L$ with minimum $L$ such that $n_l \in [k+1]$ for all $l \in [L]$.
3: **for** $l \in [L]$ **do**
4:     Initialize $b_i$ to zero for all $i \in [n]$ and $E^{(l)}$ to $\emptyset$.
5:     **for** $i \in [n]$ **do**
6:         **for** $m \in [n_l]$ **do**
7:             $j \leftarrow \mathrm{mod}(i + m \times \prod_{l'=1}^{l-1} n_{l'} - 1, n) + 1$.
8:             **if** $b_i < n_l - 1$ and $b_j < n_l - 1$ **then**
9:                 Add edge $(v_i, v_j)$ with weight $\frac{1}{n_l}$ to $E^{(l)}$ and $b_i \leftarrow b_i + 1$.
10: **return** $(E^{(1)}, E^{(2)}, \cdots, E^{(L)})$.

---

### 4.1 $k$-peer Hyper-hypercube Graph

Before proposing the BASE-$(k+1)$ GRAPH, we first extend the 1-peer hypercube graph [31] to the $k$-peer setting and propose the $k$-PEER HYPER-HYPERCUBE GRAPH, which is finite-time convergence when the number of nodes $n$ does not have prime factors larger than $k+1$ and is used as a component in the BASE-$(k+1)$ GRAPH. Let $V$ be a set of $n$ nodes. We assume that all prime factors of $n$ are less than or equal to $k+1$. That is, there exists $n_1, n_2, \cdots, n_L \in [k+1]$ such that $n = n_1 \times \cdots \times n_L$. In this case, we can construct the $L$-finite time convergent sequence of graphs whose maximum degree is at most $k$. Using Fig. 2a, we explain how all nodes reach the exact consensus. Let $G^{(1)}$ and $G^{(2)}$ denote the graphs in Fig. 2a from left to right, respectively. After the nodes exchange parameters with neighbors in $G^{(1)}$, nodes 1 and 2, nodes 3 and 4, and nodes 5 and 6 have the same parameter respectively. Then, after exchanging parameters in $G^{(2)}$, all nodes reach the exact consensus. We present the complete algorithms for constructing the $k$-PEER HYPER-HYPERCUBE GRAPH in Alg. 1.

### 4.2 Simple Base-$(k+1)$ Graph

As described in Sec. 4.1, when $n$ do not have prime factors larger than $k+1$, we can easily make all nodes reach the exact consensus by the $k$-PEER HYPER-HYPERCUBE GRAPH. However, when $n$ has prime factors larger than $k+1$, e.g., when $(k, n) = (1, 5)$, the $k$-PEER HYPER-HYPERCUBE GRAPH cannot be constructed. In this section, we extend the $k$-PEER HYPER-HYPERCUBE GRAPH and propose the SIMPLE BASE-$(k+1)$ GRAPH, which is finite-time convergence for any number of nodes $n$ and maximum degree $k$. Note that the maximum degree of the SIMPLE BASE-$(k+1)$ GRAPH is not $k+1$, but at most $k$.

We present the pseudo-code for constructing the SIMPLE BASE-$(k+1)$ GRAPH in Alg. 2. For simplicity, here we explain only the case when the maximum degree $k$ is one. The case with $k \geq 2$ is explained in Sec. B. The SIMPLE BASE-$(k+1)$ GRAPH mainly consists of the following four steps. The key idea is that splitting $V$ into disjoint subsets to which the $k$-PEER HYPER-HYPERCUBE GRAPH is applicable.

**Step 1.** As in the base-2 number of $n$, we decompose $n$ as $n = 2^{p_1} + \cdots + 2^{p_L}$ in line 1, and then split $V$ into disjoint subsets $V_1, \cdots, V_L$ such that $|V_l| = 2^{p_l}$ for all $l \in [L]$ in line 3.[3]

**Step 2.** For all $l \in [L]$, we make all nodes in $V_l$ obtain the average of parameters in $V_l$ using the 1-PEER HYPER-HYPERCUBE GRAPH $\mathcal{H}_1(V_l)$ in line 11. Then, we initialize $l'$ as one.

**Step 3.** Each node in $V_{l'+1} \cup \cdots \cup V_L$ exchanges parameters with one node in $V_{l'}$ such that the average in $V_{l'}$ becomes equivalent to the average in $V$. We increase $l'$ by one and repeat step 3 until $l' = L$. This procedure corresponds to line 15.

**Step 4.** For all $l \in [L]$, we make all nodes in $V_l$ obtain the average in $V_l$ using the 1-PEER HYPER-HYPERCUBE GRAPH $\mathcal{H}_1(V_l)$. Because the average in $V_l$ is equivalent to the average in $V$ after step 3, all nodes can reach the exact consensus. This procedure corresponds to line 25.

Using the example presented in Fig. 3, we explain Alg. 2 in more detail. Let $G^{(1)}, \cdots, G^{(5)}$ denote the graphs in Fig. 3 from left to right, respectively. In step 1, we split $V \coloneqq \{1, \cdots, 5\}$ into

---

[3]Splitting $V_l$ into $V_{l,1}, \cdots, V_{l,a_l}$ becomes crucial when $k \geq 2$ (see Sec. B).

$V_1 := \{1, \cdots, 4\}$ and $V_2 := \{5\}$. In step 2, nodes in $V_1$ obtain the average in $V_1$ by exchanging parameters in $G^{(1)}$ and $G^{(2)}$. In step 3, the average in $V_1$ becomes equivalent to the average in $V$ by exchanging parameters in $G^{(3)}$. In step 4, nodes in $V_1$ can get the average in $V$ by exchanging parameters in $G^{(4)}$ and $G^{(5)}$. Because node 5 also obtains the average in $V$ after exchanging parameters in $G^{(3)}$, all nodes reach the exact consensus after exchanging parameters in $G^{(5)}$.

Note that edges added in lines 20 and 27 are not necessary if we only need to make all nodes reach the exact consensus. Nonetheless, these edges are effective in keeping the parameters of nodes close in value to each other in decentralized learning because the parameters are updated by gradient descent before the parameter exchange with neighbors. For instance, edge $(1, 2)$ in $G^{(3)}$, which is added in line 20, is not necessary for finite-time convergence because nodes 1 and 2 already have the same parameter after exchanging parameters in $G^{(1)}$ and $G^{(2)}$. We provide more examples in Sec. C.

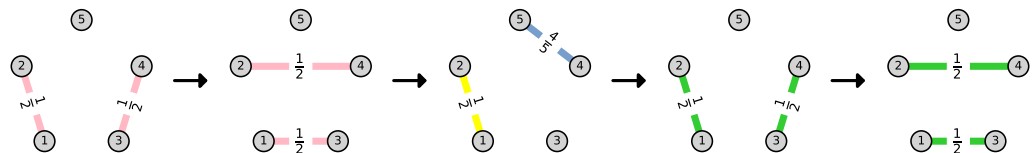

Figure 3: SIMPLE BASE-2 GRAPH with $n = 5(= 2^2 + 1)$. The value on the edge is the edge weight, and the edges are colored in the same color as the line in Alg. 2 where they were added.

---

**Algorithm 2** SIMPLE BASE-$(k + 1)$ GRAPH $\mathcal{A}_k^{\text{simple}}(V)$

---

1: **Input:** the set of nodes $V$ and number of nodes $n(= a_1(k+1)^{p_1} + a_2(k+1)^{p_2} + \cdots + a_L(k+1)^{p_L})$ such that $p_1 > p_2 > \cdots > p_L \geq 0$ and $a_l \in [k]$ for all $l \in [L]$.
2: **If** all prime factors of $n$ are less than or equal to $k + 1$ **then return** $\mathcal{H}_k(V)$.
3: Split $V$ into disjoint subsets $V_1, \cdots, V_L$ such that $|V_l| = a_l(k+1)^{p_l}$ for all $l \in [L]$. Then, for all $l \in [L]$, split $V_l$ into disjoint subsets $V_{l,1}, \cdots, V_{l,a_l}$ such that $|V_{l,a}| = (k+1)^{p_l}$ for all $a \in [a_l]$.
4: Construct $k$-PEER HYPER-HYPERCUBE GRAPH $\mathcal{H}_k(V_l)$ for all $l \in [L]$ and $m_1 = |\mathcal{H}_k(V_1)|$.
5: Construct $k$-PEER HYPER-HYPERCUBE GRAPH $\mathcal{H}_k(V_{l,a})$ for all $l \in [L]$ and $a \in [a_l]$.
6: Initialize $b_l$ as zero for all $l \in [L]$, and initialize $m$ as zero.
7: **while** $b_1 < |\mathcal{H}_k(V_{1,1})|$ **do**
8:     $m \leftarrow m + 1$ and $E^{(m)} \leftarrow \emptyset$.
9:     **for** $l \in \{L, L - 1, \cdots, 1\}$ **do**
10:         **if** $m \leq m_1$ **then**
11:             Add $E(\mathcal{H}_k(V_l)^{(m')})$ to $E^{(m)}$ where $m' = \text{mod}(m - 1, |\mathcal{H}_k(V_l)|) + 1$.
12:         **else if** $m < m_1 + l$ **then**
13:             **for** $v \in V_l$ **do**
14:                 Select isolated node $u_1, \cdots, u_{a_{m-m_1}}$ from $V_{m-m_1,1}, \cdots, V_{m-m_1,a_{m-m_1}}$.
15:                 Add edges $(v, u_1), \cdots, (v, u_{a_{m-m_1}})$ with weight $\frac{|V_{m-m_1}|}{a_{m-m_1} \sum_{l'=m-m_1}^{L} |V_{l'}|}$ to $E^{(m)}$.
16:         **else if** $m = m_1 + l$ and $l \neq L$ **then**
17:             **while** There are two or more isolated nodes in $V_l$ **do**
18:                 $c \leftarrow$ the number of isolated nodes in $V_l$.
19:                 Select $\min\{k + 1, c\}$ isolated nodes $V'$ in $V_l$.
20:                 Add edges with weights $\frac{1}{|V'|}$ to $E^{(m)}$ such that $V'$ compose the complete graph.
21:         **else**
22:             $b_l \leftarrow b_l + 1$.
23:             **if** $p_l \neq 0$ **then**
24:                 **for** $a \in [a_l]$ **do**
25:                     Add $E(\mathcal{H}_k(V_{l,a})^{(m')})$ to $E^{(m)}$ where $m' = \text{mod}(b_l - 1, |\mathcal{H}_k(V_{l,a})|) + 1$.
26:             **else**
27:                 Add $E(\mathcal{H}_k(V_l)^{(m')})$ to $E^{(m)}$ where $m' = \text{mod}(b_l - 1, |\mathcal{H}_k(V_l)|) + 1$.
28: **return** $(E^{(1)}, E^{(2)}, \cdots, E^{(m)})$.

---

**Algorithm 3** BASE-$(k+1)$ GRAPH $\mathcal{A}_k(V)$

---

1: **Input:** the set of nodes $V$ and number of nodes $n$.
2: Decompose $n$ as $n = p \times q$ such that $p$ is a multiple of $2, 3, \cdots, (k+1)$ and $q$ is prime to $2, 3, \cdots, (k+1)$.
3: Split $V$ into disjoint subsets $V_1, \cdots, V_p$ such that $|V_l| = q$ for all $l$.
4: Construct SIMPLE BASE-$(k+1)$ GRAPH $\mathcal{A}_k^{\text{simple}}(V_l)$ for all $l \in [p]$.
5: **for** $m \in \{1, 2, \cdots, |\mathcal{A}_k^{\text{simple}}(V_1)|\}$ **do**
6: $\quad E^{(m)} \leftarrow \bigcup_{l \in [p]} E(\mathcal{A}_k^{\text{simple}}(V_l)^{(m)})$.
7: Split $V$ into disjoint subsets $U_1, \cdots, U_q$ such that $|U_l| = p$ and $|U_l \cap V_{l'}| = 1$ for all $l, l'$.
8: Construct $k$-PEER HYPER-HYPERCUBE GRAPH $\mathcal{H}_k(U_l)$ for all $l \in [q]$.
9: **for** $m \in \{1, 2, \cdots, |\mathcal{H}_k(U_1)|\}$ **do**
10: $\quad E^{(m+|\mathcal{A}_k^{\text{simple}}(V_1)|)} \leftarrow \bigcup_{l \in [q]} E(\mathcal{H}_k(U_l)^{(m)})$.
11: $\mathcal{E} \leftarrow (E^{(1)}, E^{(2)}, \cdots, E^{(|\mathcal{A}_k^{\text{simple}}(V_1)| + |\mathcal{H}_k(U_1)|)})$.
12: **If** $|\mathcal{A}_k^{\text{simple}}(V)| < |\mathcal{E}|$ **then return** $\mathcal{A}_k^{\text{simple}}(V)$ **else return** $\mathcal{E}$.

---

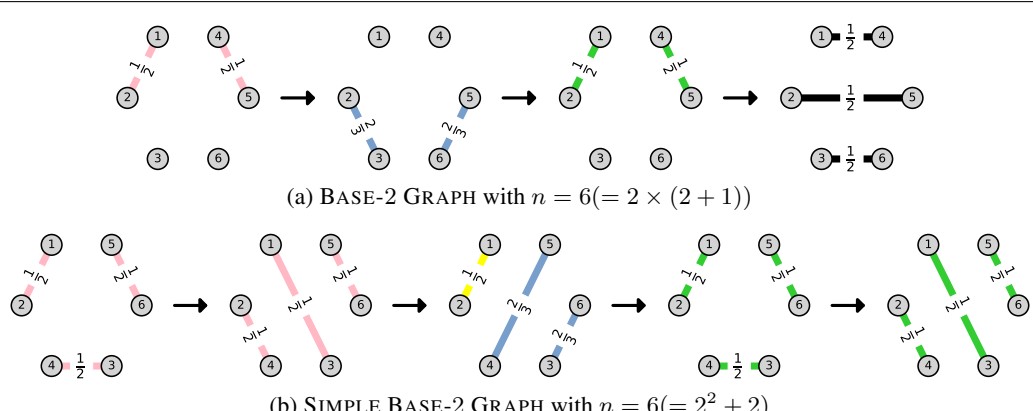

(a) BASE-2 GRAPH with $n = 6 (= 2 \times (2+1))$

(b) SIMPLE BASE-2 GRAPH with $n = 6 (= 2^2 + 2)$

Figure 4: Comparison of SIMPLE BASE-2 GRAPH and BASE-2 GRAPH with $n = 6$. The value on the edge indicates the edge weight. The edges added in line 10 in Alg. 3 are colored black, and the edges added in line 6 are colored the same color as the line in Alg. 2 where they are added.

### 4.3 Base-$(k+1)$ Graph

The SIMPLE BASE-$(k+1)$ GRAPH is finite-time convergence for any $n$ and $k$, while the SIMPLE BASE-$(k+1)$ GRAPH contains graphs that are not necessary for the finite-time convergence and becomes a redundant sequence of graphs in some cases, e.g., $(k, n) = (1, 6)$ (see the example in Fig. 4b and a detailed explanation in Sec. C.2). To remove this redundancy, this section proposes the BASE-$(k+1)$ GRAPH that can make all nodes reach the exact consensus after fewer iterations than the SIMPLE BASE-$(k+1)$ GRAPH.

The pseudo-code for constructing the BASE-$(k+1)$ GRAPH is shown in Alg. 3. The BASE-$(k+1)$ GRAPH consists of the following three steps.

**Step 1.** We decompose $n$ as $p \times q$ such that $p$ is a multiple of $2, \cdots, (k+1)$ and $q$ is prime to $2, \cdots, (k+1)$, and split $V$ into disjoint subsets $V_1, \cdots, V_p$ such that $|V_l| = q$ for all $l \in [p]$.

**Step 2.** For all $l \in [p]$, we make all nodes in $V_l$ reach the average in $V_l$ by the SIMPLE BASE-$(k+1)$ GRAPH $\mathcal{A}_k^{\text{simple}}(V_l)$. Then, we take $p$ nodes from $V_1, \cdots, V_p$ respectively and construct a set $U_1$. Similarly, we construct $U_2, \cdots, U_q$ such that $U_1, \cdots, U_q$ are disjoint sets.

**Step 3.** For all $l \in [q]$, we make all nodes in $U_l$ reach the average in $U_l$ by the $k$-PEER HYPER-HYPERCUBE GRAPH $\mathcal{H}_k(U_l)$. Because the average in $U_l$ is equivalent to the average in $V$ after step 2, all nodes reach the exact consensus.

Using the example in Fig. 4a, we explain the BASE-$(k+1)$ GRAPH in more detail. Let $G^{(1)}, \cdots, G^{(4)}$ denote the graphs in Fig. 4a from left to right, respectively. In step 1, we split $V$ into $V_1 \coloneqq \{1, 2, 3\}$ and $V_2 \coloneqq \{4, 5, 6\}$. In step 2, nodes in $V_1$ and nodes in $V_2$ have the same parameter respectively by

exchanging parameters on $G^{(1)}, \cdots, G^{(3)}$ because the subgraphs composed on $V_1$ and $V_2$ are same as the SIMPLE BASE-$(k+1)$ GRAPH (see Fig. 14a). Then, we construct $U_1 := \{1, 4\}$, $U_2 := \{2, 5\}$, and $U_3 := \{3, 6\}$. Finally, in step 3, all nodes reach the exact consensus by exchanging parameters in $G^{(4)}$.

Fig. 5 and Sec. F.1 compare the BASE-$(k+1)$ GRAPH with the SIMPLE BASE-$(k+1)$ GRAPH, demonstrating that the length of the BASE-$(k+1)$ GRAPH is less than that of the SIMPLE BASE-$(k+1)$ GRAPH in many cases. Moreover, Theorem 1 show the upper bound of the length of the SIMPLE BASE-$(k+1)$ GRAPH and BASE-$(k+1)$ GRAPH. The proof is provided in Sec. D.

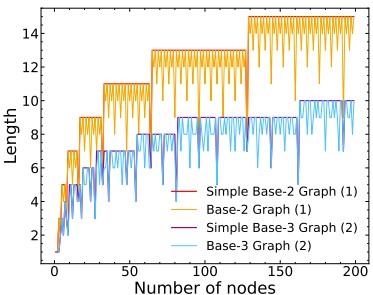

Figure 5: Comparison of length.

**Theorem 1.** *For any number of nodes $n \in \mathbb{N}$ and maximum degree $k \in [n-1]$, the length of the SIMPLE BASE-$(k+1)$ GRAPH and BASE-$(k+1)$ GRAPH is less than or equal to $2\log_{k+1}(n) + 2$.*

**Corollary 1.** *For any number of nodes $n \in \mathbb{N}$ and maximum degree $k \in [n-1]$, the SIMPLE BASE-$(k+1)$ GRAPH and BASE-$(k+1)$ GRAPH are $\mathcal{O}(\log_{k+1}(n))$-finite time convergence.*

Therefore, the BASE-$(k+1)$ GRAPH is a powerful extension of the 1-peer exponential [43] and 1-peer hypercube graphs [31] because they are $\mathcal{O}(\log_2(n))$-finite time convergence only if $n$ is a power of 2 and their maximum degree cannot be set to any number other than 1.

## 5  Decentralized SGD on Base-$(k+1)$ Graph

In this section, we verify the effectiveness of the BASE-$(k+1)$ GRAPH for decentralized learning, demonstrating that the BASE-$(k+1)$ GRAPH can endow DSGD with both a faster convergence rate and fewer communication costs than the existing topologies, including the ring, torus, and exponential graph. We consider the following decentralized learning problem:

$$\min_{\boldsymbol{x} \in \mathbb{R}^d} \left[ f(\boldsymbol{x}) := \frac{1}{n} \sum_{i=1}^{n} f_i(\boldsymbol{x}) \right], \quad f_i(\boldsymbol{x}) := \mathbb{E}_{\xi_i \sim \mathcal{D}_i} \left[ F_i(\boldsymbol{x}; \xi_i) \right],$$

where $n$ is the number of nodes, $f_i$ is the loss function of node $i$, $\mathcal{D}_i$ is the data distribution held by node $i$, $F_i(\boldsymbol{x}; \xi_i)$ is the loss of node $i$ at data sample $\xi_i$, and $\nabla F_i(\boldsymbol{x}; \xi_i)$ denotes the stochastic gradient. Then, we assume that the following hold, which are commonly used for analyzing decentralized learning methods [18, 20, 23, 43].

**Assumption 1.** *There exists $f^\star > -\infty$ that satisfies $f(\boldsymbol{x}) \geq f^\star$ for any $\boldsymbol{x} \in \mathbb{R}^d$.*

**Assumption 2.** *$f_i$ is $L$-smooth for all $i \in [n]$.*

**Assumption 3.** *There exists $\sigma^2$ that satisfies $\mathbb{E}_{\xi_i \sim \mathcal{D}_i} \|\nabla F_i(\boldsymbol{x}; \xi_i) - \nabla f_i(\boldsymbol{x})\|^2 \leq \sigma^2$ for all $\boldsymbol{x} \in \mathbb{R}^d$.*

**Assumption 4.** *There exists $\zeta^2$ that satisfies $\frac{1}{n} \sum_{i=1}^{n} \|\nabla f_i(\boldsymbol{x}) - \nabla f(\boldsymbol{x})\|^2 \leq \zeta^2$ for all $\boldsymbol{x} \in \mathbb{R}^d$.*

We consider the case when DSGD [18], the most widely used decentralized learning method, is used as an optimization method. Let $\boldsymbol{W}^{(1)}, \cdots, \boldsymbol{W}^{(m)}$ be mixing matrices of the BASE-$(k+1)$ GRAPH. In DSGD on the BASE-$(k+1)$ GRAPH, node $i$ updates its parameter $\boldsymbol{x}_i$ as follows:

$$\boldsymbol{x}_i^{(r+1)} = \sum_{j=1}^{n} W_{ij}^{(1+\text{mod}(r,m))} \left( \boldsymbol{x}_j^{(r)} - \eta \nabla F_j(\boldsymbol{x}_j^{(r)}; \xi_j^{(r)}) \right), \tag{1}$$

where $\eta$ is the learning rate. In this case, thanks to the property of finite-time convergence, DSGD on the BASE-$(k+1)$ GRAPH converges at the following convergence rate.

**Theorem 2.** *Suppose that Assumptions 1-4 hold. Then, for any number of nodes $n \in \mathbb{N}$ and maximum degree $k \in [n-1]$, there exists $\eta$ such that $\bar{\boldsymbol{x}} := \frac{1}{n} \sum_{i=1}^{n} \boldsymbol{x}_i$ generated by Eq. (1) satisfies $\frac{1}{R+1} \sum_{r=0}^{R} \mathbb{E} \left\| \nabla f(\bar{\boldsymbol{x}}^{(r)}) \right\|^2 \leq \epsilon$ after*

$$R = \mathcal{O} \left( \frac{\sigma^2}{n\epsilon^2} + \frac{\zeta \log_{k+1}(n) + \sigma \sqrt{\log_{k+1}(n)}}{\epsilon^{3/2}} + \frac{\log_{k+1}(n)}{\epsilon} \right) \cdot L F_0 \tag{2}$$

*iterations, where $F_0 := f(\bar{\boldsymbol{x}}^{(0)}) - f^\star$.*

The above theorem follows immediately from Theorem 2 stated in Koloskova et al. [11] and Corollary 1. The convergence rates of DSGD over commonly used topologies are summarized in Sec. E. From Theorem 2 and Sec. E, we can conclude that for any number of nodes $n$, the BASE-2 GRAPH enables DSGD to converge faster than the ring and torus and as fast as the exponential graph, although the maximum degree of the BASE-2 GRAPH is only one. Moreover, if we set the maximum degree $k$ to the value between 2 to $\lceil \log_2(n) \rceil$, the BASE-$(k+1)$ GRAPH enables DSGD to converge faster than the exponential graph, even though the maximum degree of the BASE-$(k+1)$ GRAPH remains less than that of the exponential graph. It is worth noting that if we increase the maximum degree of the 1-peer exponential and 1-peer hypercube graphs (i.e., $k$-peer exponential and $k$-peer hypercube graphs with $k \geq 2$), these topologies cannot enable DSGD to converge faster than the exponential graph because these topologies are no longer finite-time convergence even when the number of nodes is a power of 2.

## 6 Experiments

In this section, we validate the effectiveness of the BASE-$(k+1)$ GRAPH. First, we experimentally verify that the BASE-$(k+1)$ GRAPH is finite-time convergence for any number of nodes in Sec. 6.1, and we verify the effectiveness of the BASE-$(k+1)$ GRAPH for decentralized learning in Sec. 6.2.

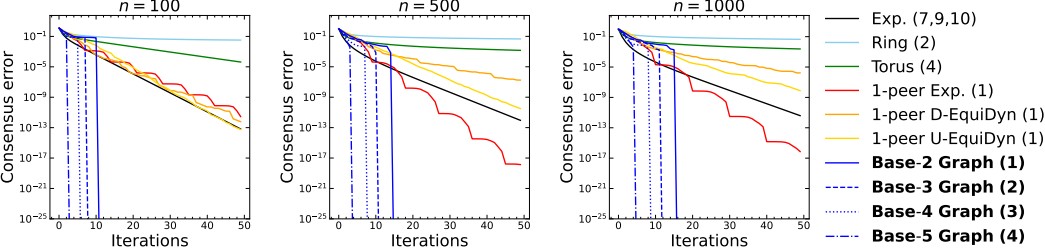

Figure 6: Comparison of consensus rates among various topologies. The number in the bracket indicates the maximum degree of a topology. Because the maximum degree of the exponential graph depends on $n$, the three numbers in the bracket indicate the maximum degree for each $n$.

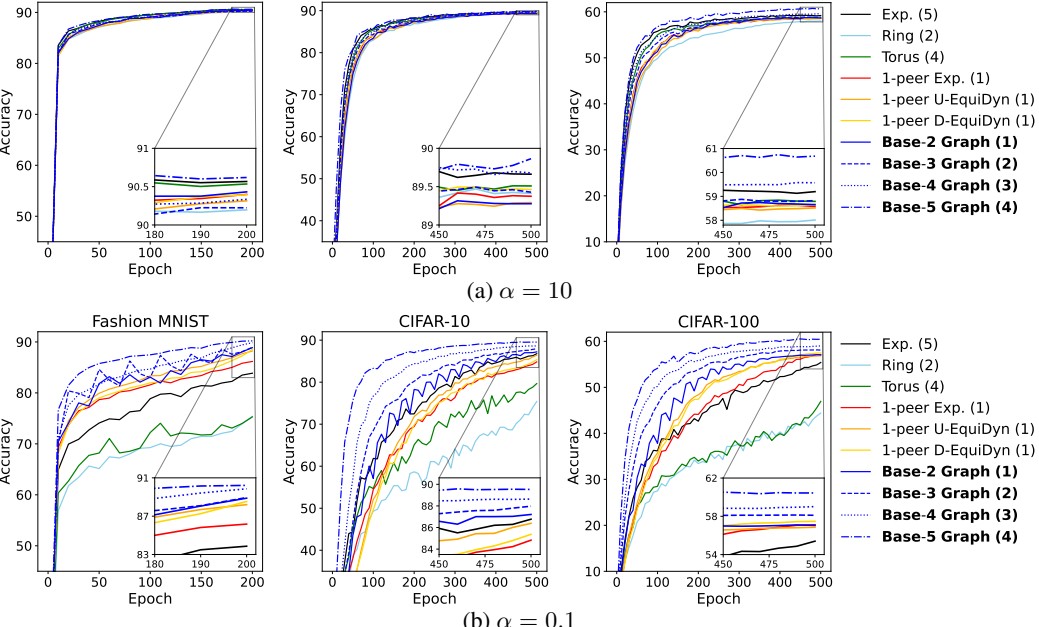

Figure 7: Test accuracy (%) of DSGD on various topologies with $n = 25$. The number in the bracket indicates the maximum degree of a topology. We also compared with dense variants of the 1-peer {U, D}-EquiDyn [33] in Sec. F.3.1, showing the superior performance of the BASE-$(k+1)$ GRAPH.

### 6.1 Consensus Rate

**Setup.** Let $x_i \in \mathbb{R}$ be the parameter that node $i$ has, and let $\bar{x} := \frac{1}{n} \sum_{i=1}^{n} x_i$. For each $i$, the initial value of $x_i$ was drawn from Gaussian distribution with mean $0$ and standard variance $1$. Then, we evaluated how the consensus error $\frac{1}{n} \sum_{i=1}^{n} (x_i - \bar{x})^2$ decreases when $x_i$ is updated as $x_i \leftarrow \sum_{j=1}^{n} W_{ij} x_j$ where $\boldsymbol{W}$ is the mixing matrix associated with a given topology.

**Results.** Figs. 1 and 6 present how the consensus errors decrease on various topologies. The results indicate that the BASE-$(k+1)$ GRAPH reaches the exact consensus after a finite number of iterations, while the other topologies only reach the consensus asymptotically. Moreover, as the maximum degree $k$ increases, the BASE-$(k+1)$ GRAPH reaches the exact consensus with fewer iterations. We also present the results when $n$ is a power of 2 in Sec. F.2, demonstrating that the 1-peer exponential graph can reach the exact consensus as well as the BASE-2 GRAPH, but requires more iterations than the BASE-4 GRAPH.

### 6.2 Decentralized Learning

Next, we examine the effectiveness of the BASE-$(k+1)$ GRAPH in decentralized learning.

**Setup.** We used three datasets, Fashion MNIST [41], CIFAR-$\{10, 100\}$ [14], and used LeNet [15] for Fashion MNIST and VGG-11 [32] for CIFAR-$\{10, 100\}$. Additionally, we present the results using ResNet-18 [5] in Sec. G. The learning rate was tuned by the grid search and we used the cosine learning rate scheduler [22]. We distributed the training dataset to nodes by using Dirichlet distributions with hyperparameter $\alpha$ [7], conducting experiments in both homogeneous and heterogeneous data distribution settings. As $\alpha$ approaches zero, the data distributions held by each node become more heterogeneous. We repeated all experiments with three different seed values and reported their averages. See Sec. H for more detailed settings.

**Results of DSGD on Various Topologies.** We compared various topologies combined with the DSGD with momentum [4, 18], showing the results in Fig. 7. From Fig. 7, the accuracy differences among topologies are larger as the data distributions are more heterogeneous. From Fig. 7b, the BASE-$\{2, 3, 4, 5\}$ GRAPH reach high accuracy faster than the other topologies. Furthermore, comparing the final accuracy, the final accuracy of the BASE-2 GRAPH is comparable to or higher than that of the existing topologies, including the exponential graph.

Moreover, the final accuracy of the BASE-$\{3, 4, 5\}$ GRAPH is higher than that of all existing topologies. From Fig. 7a, the accuracy differences among topologies become small when $\alpha = 10$; however, the BASE-5 GRAPH still outperforms the other topologies. In Fig. 8, we present the results in cases other than $n = 25$, demonstrating that the BASE-2 GRAPH outperforms the 1-peer exponential graph and the BASE-$\{3, 4, 5\}$ GRAPH can consistently outperform the exponential and 1-peer exponential graphs for all $n$. In Sec. F.3.2, we show the learning curves and the comparison of the consensus rate when $n$ is 21, 22, 23, 24, and 25.

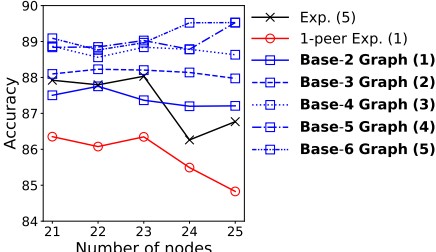

Figure 8: Test accuracy (%) of DSGD with CIFAR-10 and $\alpha = 0.1$.

**Results of $D^2$ and QG-DSGDm on Various Topologies.** The above results demonstrated that the BASE-$(k+1)$ GRAPH outperforms the existing topologies, especially when the data distributions are heterogeneous. Hence, we next compared the BASE-$(k+1)$ GRAPH with the existing topologies in the case where $D^2$ [36] and QG-DSGDm [20], which are robust to data heterogeneity, are used as decentralized learning methods. From Fig. 9, the BASE-2 GRAPH can achieve comparable or higher accuracy than the 1-peer exponential graph, and the BASE-5 GRAPH consistently out-

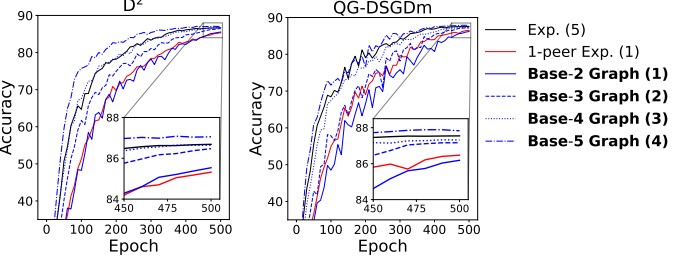

Figure 9: Test accuracy (%) of $D^2$ and QG-DSGDm with CIFAR-10, $n = 25$, and $\alpha = 0.1$.

performs the exponential graph. Thus, the BASE-$(k+1)$ GRAPH is useful not only for DSGD but also for $D^2$ and QG-DSGDm and then enables these methods to achieve a reasonable balance between accuracy and communication efficiency.

# 7 Conclusion

In this study, we propose the BASE-$(k + 1)$ GRAPH, a novel topology with both a fast consensus rate and small maximum degree. Unlike the existing topologies, the BASE-$(k + 1)$ GRAPH is finite-time convergence for any number of nodes and maximum degree $k$. Thanks to this favorable property, the BASE-$(k + 1)$ GRAPH enables DSGD to obtain both a faster convergence rate and more communication efficiency than the existing topologies, including the ring, torus, and exponential graph. Through experiments, we compared the BASE-$(k+1)$ GRAPH with various existing topologies, demonstrating that the BASE-$(k+1)$ GRAPH enables various decentralized learning methods to more successfully reconcile accuracy and communication efficiency than the existing topologies.

## Acknowledgments

Yuki Takezawa, Ryoma Sato, and Makoto Yamada were supported by JSPS KAKENHI Grant Number 23KJ1336, 21J22490, and MEXT KAKENHI Grant Number 20H04243, respectively.

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

# A  Detailed Explanation of $k$-peer Hyperhypercube Graph

In this section, we explain Alg. 1 in more detail. The $k$-PEER HYPER-HYPERCUBE GRAPH mainly consists of the following five steps.

**Step 1.** Decompose $n$ as $n = n_1 \times \cdots \times n_L$ with minimum $L$ such that $n_l \in [k+1]$ for all $l \in [L]$.

**Step 2.** If $L = 1$, we make all nodes obtain the average of parameters in $V$ by using the complete graph. If $L \geq 2$, we split $V$ into disjoint subsets $V_1, \cdots, V_{n_L}$ such that $|V_l| = \frac{n}{n_L}$ for all $l \in [n_L]$ and continue to step 3.

**Step 3.** For all $l \in [n_L]$, we make all nodes in $V_l$ obtain the average of parameters in $V_l$ by using the $k$-PEER HYPER-HYPERCUBE GRAPH $\mathcal{H}_k(V_l)$.

**Step 4.** We take $n_L$ nodes from $V_1, \cdots, V_{n_L}$ respectively and construct a set $U_1$. Similarly, we construct $U_2, \cdots, U_{n_L}$ such that $U_1, \cdots, U_{n_L}$ are disjoint sets.

**Step 5.** For all $l \in [n_L]$, we make all nodes in $U_l$ obtain the average of parameters in $U_l$ by using the complete graph. Because the average of parameter $U_l$ is equivalent to the average in $V$ after step 4, all nodes reach the exact consensus.

When $n \leq k+1$, the $k$-PEER HYPER-HYPERCUBE GRAPH becomes the complete graph because of step 2. When $n > k+1$, we decompose $n$ in step 1 and construct the $k$-PEER HYPER-HYPERCUBE GRAPH recursively in step 3. Thus, the $k$-PEER HYPER-HYPERCUBE GRAPH can make all nodes reach the exact consensus by the sequence of $L$ graphs.

Using the example provided in Fig. 10, we explain the $k$-PEER HYPER-HYPERCUBE GRAPH in a more detailed manner. When $n = 12$, we decompose 12 as $2 \times 2 \times 3$. In step 2, we split $V := \{1, \cdots, 12\}$ into $V_1 := \{1, \cdots, 4\}$, $V_2 := \{5, \cdots, 8\}$, and $V_3 := \{9, \cdots, 12\}$. Step 3 corresponds to the first two graphs in Fig. 10b. As shown in Fig. 10a, the subgraphs consisting of $V_1$, $V_2$, and $V_3$ in the first two graphs in Fig. 10b are equivalent to the $k$-PEER HYPER-HYPERCUBE GRAPH with the number of nodes 4. Thus, all nodes reach the exact consensus by exchanging parameters in Fig. 10b.

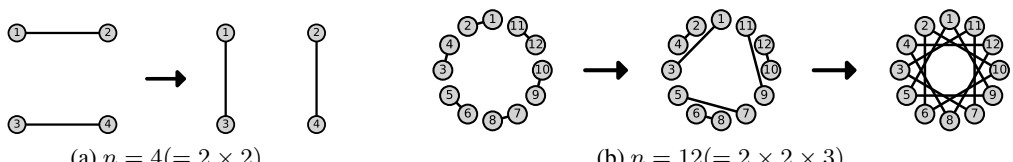

(a) $n = 4 (= 2 \times 2)$           (b) $n = 12 (= 2 \times 2 \times 3)$

Figure 10: Illustration of the 2-PEER HYPER-HYPERCUBE GRAPH. In Fig. 10a, all edge weights are $\frac{1}{2}$. In Fig. 10b, edge weights are $\frac{1}{2}$ in the first two graphs and $\frac{1}{3}$ in the last graph.

# B    Detailed Explanation of Simple Base-$(k+1)$ Graph with $k \geq 2$

In Sec. 4.2, we explain Alg. 2 only in the case where maximum degree $k$ is one. In this section, we explain the details of Alg. 2 in the case with $k \geq 2$.

The SIMPLE BASE-$(k+1)$ GRAPH mainly consists of the following five steps.

**Step 1.** As in the base-$(k+1)$ number of $n$, we decompose $n$ as $n = a_1(k+1)^{p_1} + \cdots + a_L(k+1)^{p_L}$ in line 1, and then split $V$ into disjoint subsets $V_1, \cdots, V_L$ such that $|V_l| = a_l(k+1)^{p_l}$ for all $l \in [L]$.

**Step 2.** For all $l \in [L]$, we split $V_l$ into disjoint subsets $V_{l,1}, \cdots, V_{l,a_l}$ such that $|V_{l,a}| = (k+1)^{p_l}$ for all $a \in [a_l]$ in line 3.

**Step 3.** For all $l \in [L]$, we make all nodes in $V_l$ obtain the average of parameters in $V_l$ using the $k$-PEER HYPER-HYPERCUBE GRAPH $\mathcal{H}_k(V_l)$ in line 11. Then, we initialize $l'$ as one.

**Step 4.** Each node in $V_{l'+1} \cup \cdots \cup V_L$ exchange parameters with $a_{l'}$ nodes in $V_{l'} (= V_{l',1} \cup \cdots \cup V_{l',a_{l'}})$ such that the average in $V_{l',a}$ becomes equivalent to the average in $V$ for all $a \in [a_{l'}]$. We increase $l'$ by one and repeat step 4 until $l' = L$. This procedure corresponds to line 15.

**Step 5.** For all $l \in [L]$ and $a \in [a_l]$, we make all nodes in $V_{l,a}$ obtain the average in $V_{l,a}$ using the $k$-PEER HYPER-HYPERCUBE GRAPH $\mathcal{H}_k(V_{l,a})$. Since the average in $V_{l,a}$ is equivalent to the average in $V$ after step 4, all nodes reach the exact consensus. This procedure corresponds to line 25.

The major difference compared with the case where $k = 1$ is step 4. In the case where $k = 1$, each node in $V_{l'+1} \cup \cdots \cup V_L$ exchange parameters with one node in $V_{l'}$ such that the average in $V_{l'}$ becomes equivalent to that in $V$, while in the case where $k \geq 2$, each node in $V_{l'+1} \cup \cdots \cup V_L$ exchange parameters such that the average in $V_{l',a}$ becomes equivalent to that in $V$ for all $a \in [a_l]$. Thanks to this step, we can make all nodes reach the exact consensus using $k$-PEER HYPER-HYPERCUBE GRAPH $\mathcal{H}_k(V_{l,a})$ instead of $\mathcal{H}_k(V_l)$ in step 5, and we can reduce the length of a graph sequence.

Using the example provided in Fig. 11, we explain Alg. 2 in a more detailed manner. Let $G^{(1)}, \cdots, G^{(4)}$ denote the graphs depicted in Fig. 11 from left to right, respectively. First, we split $V := \{1, \cdots, 7\}$ into $V_1 := \{1, \cdots, 6\}$ and $V_2 := \{7\}$, and then split $V_1$ into $V_{1,1} := \{1, 2, 3\}$ and $V_{1,2} := \{4, 5, 6\}$. In step 3, all nodes in $V_1$ obtain the same parameter by exchanging parameters in $G^{(1)}$ and $G^{(2)}$. In step 4, the average in $V_{1,1}$, that in $V_{1,2}$, and that in $V_2$ become the same as the average in all nodes $V$ by exchanging parameters in $G^{(3)}$. Thus, in step 5, all nodes reach the exact consensus by exchanging parameters in $G^{(4)}$.

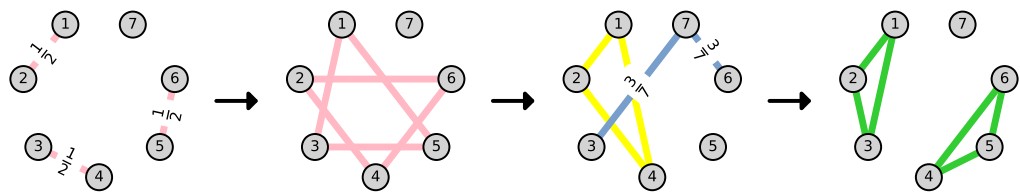

Figure 11: $k = 2, n = 7(= 2 \times 3 + 1)$. The value on the edge indicates the edge weight. For simplicity, we omit the edge value when it is $\frac{1}{3}$.

# C Illustration of Topologies

## C.1 Examples

Fig. 12 shows the examples of the SIMPLE BASE-$(k+1)$ GRAPH. Using these examples, we explain how all nodes reach the exact consensus.

We explain the case depicted in Fig. 12a. Let $G^{(1)}, G^{(2)}, G^{(3)}$ denote the graphs depicted in Fig. 12a from left to right, respectively. First, we split $V := \{1, \cdots, 5\}$ into $V_1 := \{1, 2, 3\}$ and $V_2 := \{4, 5\}$, and then split $V_2$ into $V_{2,1} := \{4\}$ and $V_{2,2} := \{5\}$. After exchanging parameters in $G^{(1)}$, nodes in $V_1$ and nodes in $V_2$ have the same parameter respectively. Then, after exchanging parameters in $G^{(2)}$, the average in $V_1$, that in $V_{2,1}$, and that in $V_{2,2}$ become same as the average in $V$. Thus, by exchanging parameters in $G^{(3)}$, all nodes reach the exact consensus. Note that edge $(4, 5)$ in $G^{(3)}$, which is added in line 27 in Alg. 2, is not necessary for all nodes to reach the exact consensus because nodes 4 and 5 already have the same parameter after exchanging parameters in $G^{(2)}$; however, it is effective in decentralized learning as we explained in Sec. 4.2.

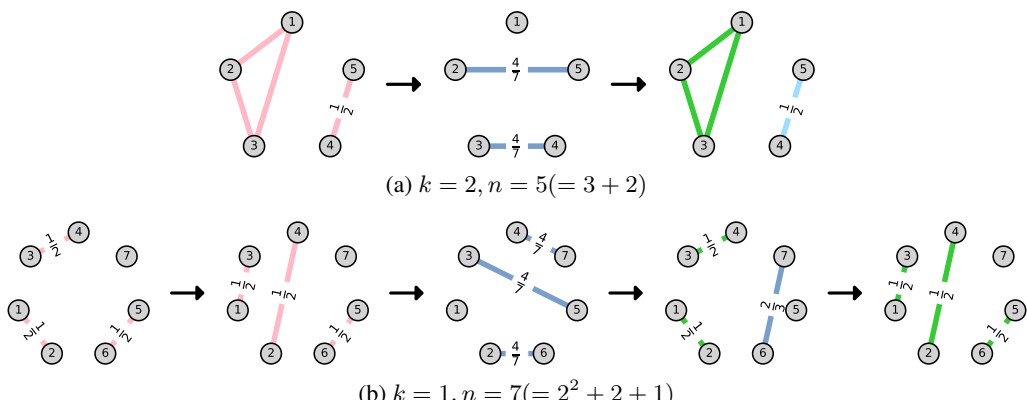

(a) $k = 2, n = 5 (= 3 + 2)$

(b) $k = 1, n = 7 (= 2^2 + 2 + 1)$

Figure 12: Illustration of the SIMPLE BASE-$(k + 1)$ GRAPH. The edge is colored in the same color as the line of Alg. 2 where the edge is added. The value on the edge indicates the edge weight. For simplicity, we omit the edge value when it is $\frac{1}{3}$.

## C.2 Illustrative Comparison between Simple Base-$(k + 1)$ and Base-$(k + 1)$ Graphs

In this section, we provide an example of the SIMPLE BASE-$(k + 1)$ GRAPH, explaining the reason why the length of the BASE-$(k + 1)$ GRAPH is less than that of the SIMPLE BASE-$(k + 1)$ GRAPH.

Let $G^{(1)}, \cdots, G^{(5)}$ denote the graphs depicted in Fig. 13 from left to right, respectively. $(G^{(1)}, G^{(2)}, G^{(3)}, G^{(4)}, G^{(5)})$ is finite-time convergence, but $(G^{(1)}, G^{(2)}, G^{(3)}, G^{(5)})$ is also finite-time convergence because after exchanging parameters in $G^{(3)}$, nodes 3 and 4 already have the same parameters. Then, using the technique proposed in Sec. 4.3, we can remove such unnecessary graphs contained in the SIMPLE BASE-$(k + 1)$ GRAPH (see Fig. 4a). Consequently, the BASE-$(k + 1)$ GRAPH can make all nodes reach the exact consensus faster than the SIMPLE BASE-$(k + 1)$ GRAPH.

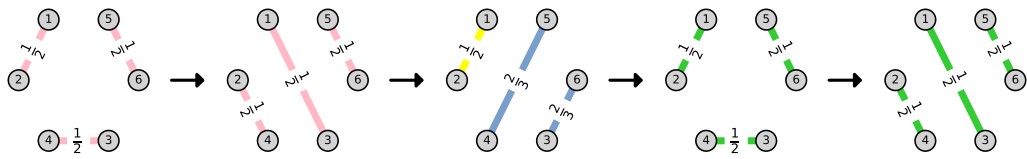

Figure 13: Illustration of the SIMPLE BASE-2 GRAPH with $n = 6 (= 2^2 + 2)$. The edge is colored in the same color as the line of Alg. 2 where the edge is added.

## C.3 Additional Examples

### C.3.1 Simple Base-$(k + 1)$ Graph

(a) $n = 3$

(b) $n = 4$

(c) $n = 5$

(d) $n = 6$

(e) $n = 7$

(f) $n = 8$

(g) $n = 9$

(h) $n = 10$

Figure 14: Illustration of the SIMPLE BASE-2 GRAPH with the various numbers of nodes.

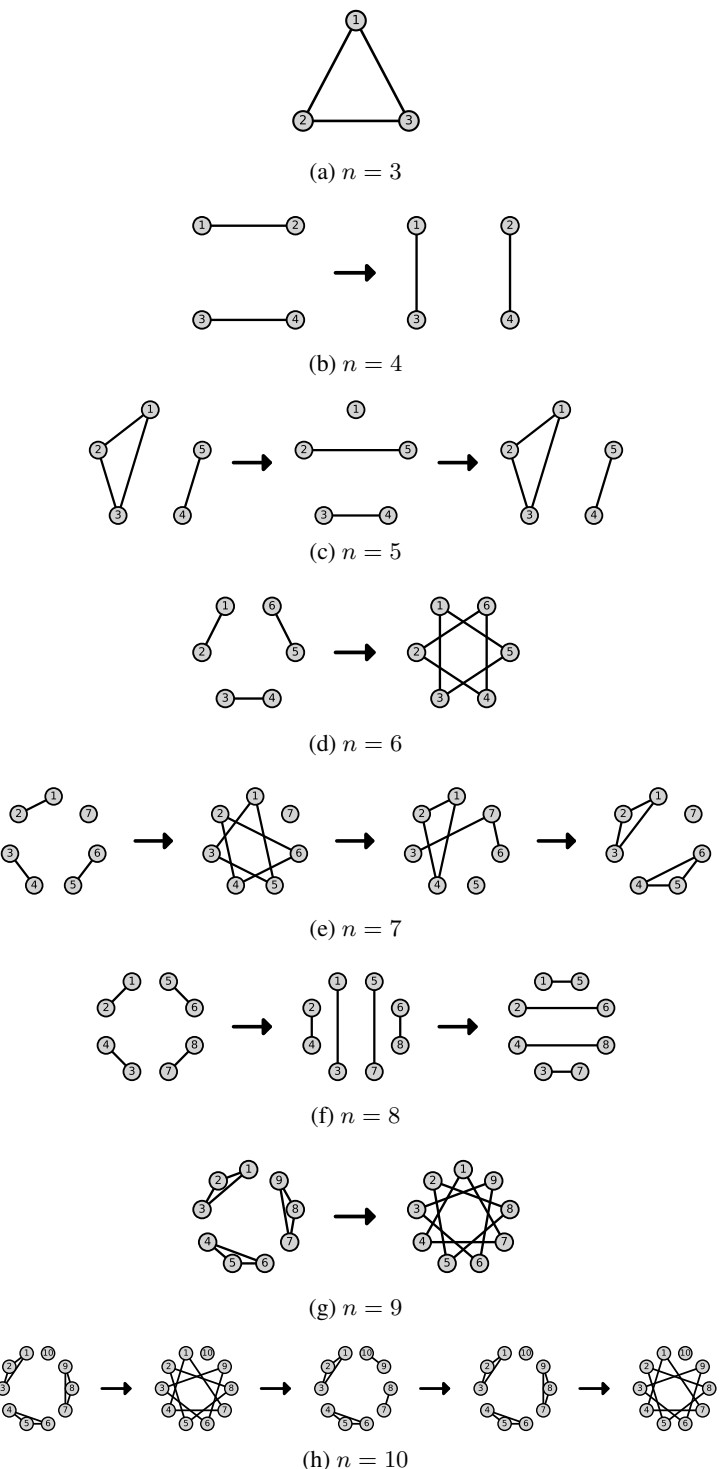

Figure 15: Illustration of the SIMPLE BASE-3 GRAPH with the various numbers of nodes.

## C.3.2 Base-$(k+1)$ Graph

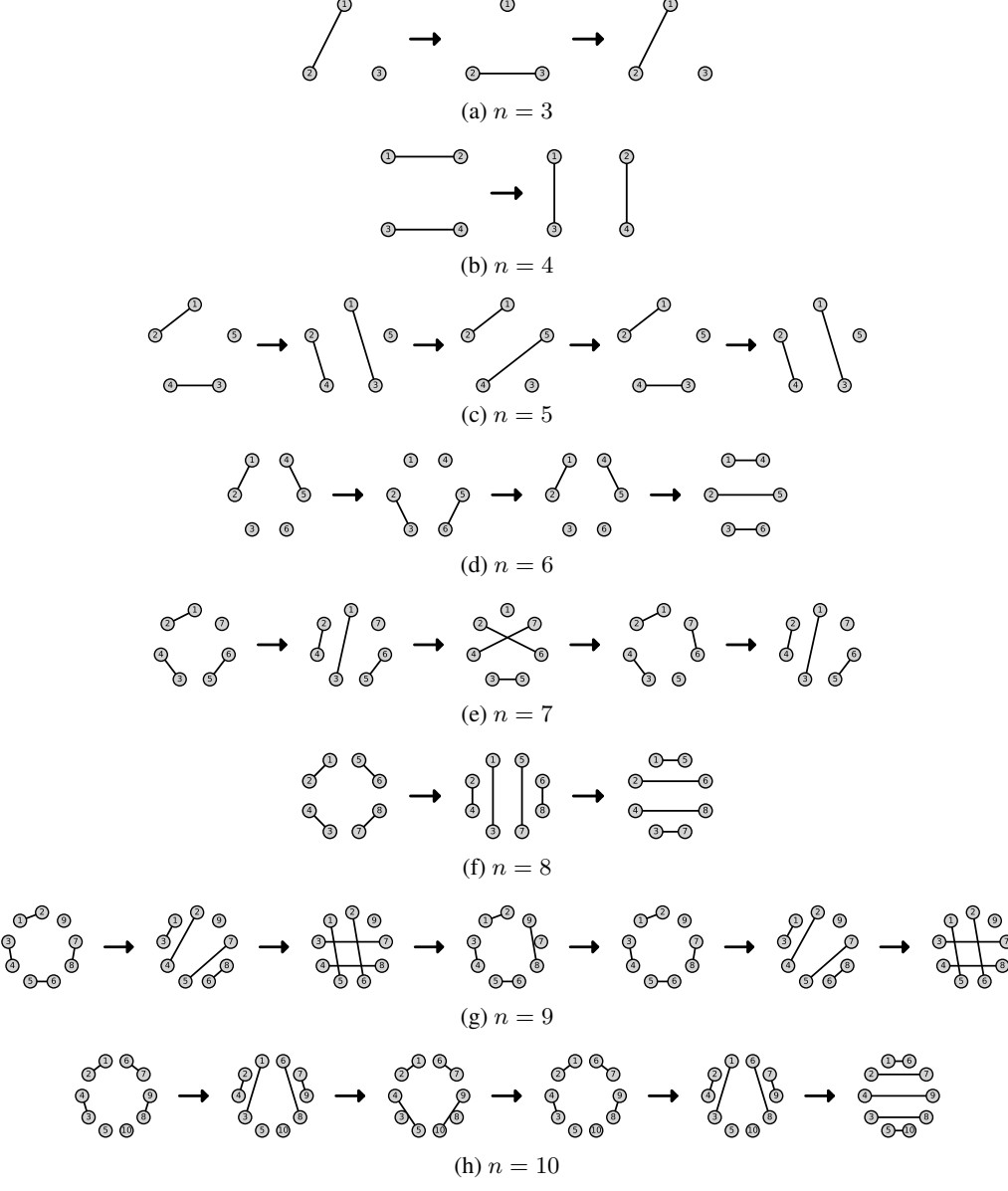

(a) $n = 3$

(b) $n = 4$

(c) $n = 5$

(d) $n = 6$

(e) $n = 7$

(f) $n = 8$

(g) $n = 9$

(h) $n = 10$

Figure 16: Illustration of the BASE-2 GRAPH with the various numbers of nodes.

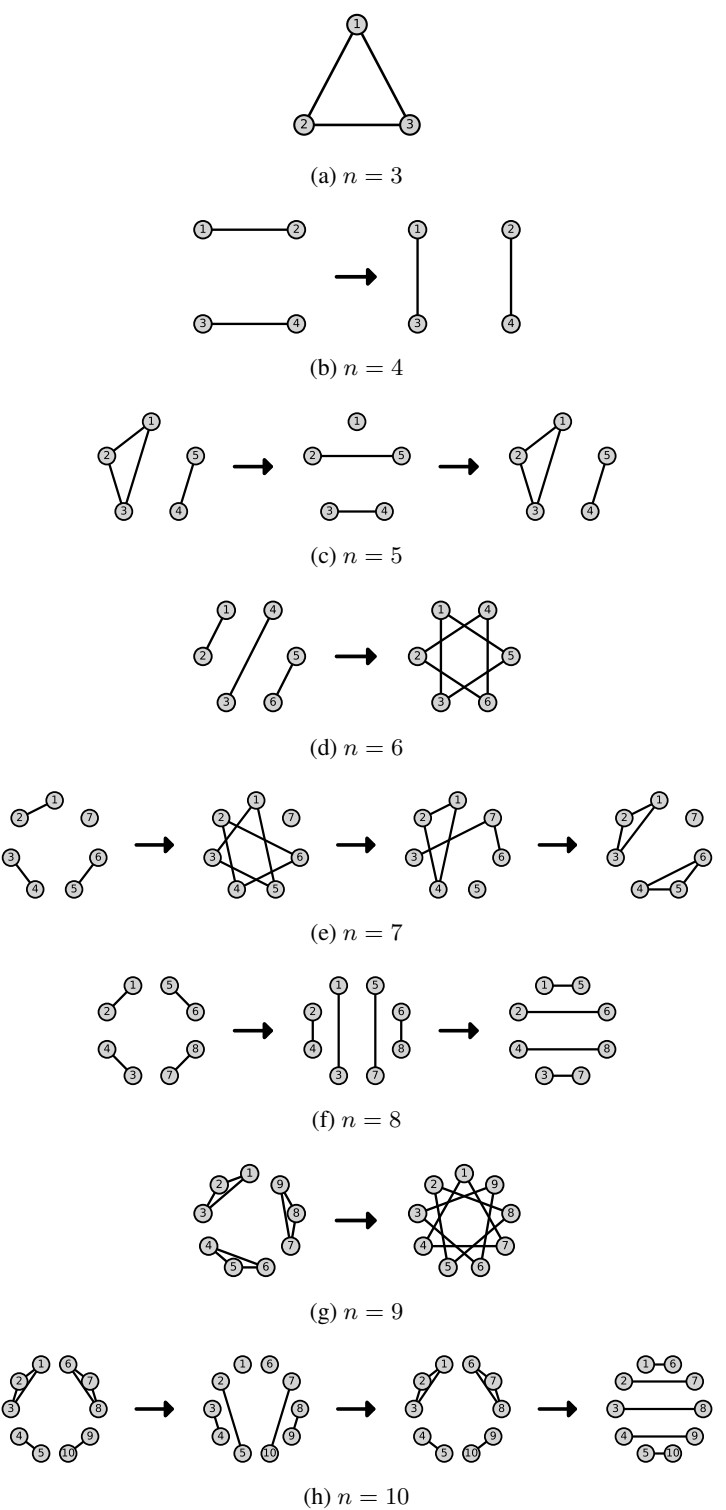

(a) $n = 3$

(b) $n = 4$

(c) $n = 5$

(d) $n = 6$

(e) $n = 7$

(f) $n = 8$

(g) $n = 9$

(h) $n = 10$

Figure 17: Illustration of the BASE-3 GRAPH with the various numbers of nodes.

## C.4 1-peer Hypercube Graph and 1-peer Exponential Graph

For completeness, we provide examples of the 1-peer hypercube [31] and 1-peer exponential graphs [43] in Figs. 19 and 18, respectively.

(a) $n = 4$

(b) $n = 8$

Figure 18: Illustration of the 1-peer hypercube graph. All edge weights are $0.5$.

(a) $n = 4$

(b) $n = 5$

(c) $n = 6$

(d) $n = 7$

(e) $n = 8$

Figure 19: Illustration of the 1-peer exponential graph. All edge weights are $0.5$.

# D  Proof of Theorem 1

**Lemma 1** (Length of $k$-PEER HYPER-HYPERCUBE GRAPH). *Suppose that all prime factors of the number of nodes $n$ are less than or equal to $k+1$. Then, for any number of nodes $n \in \mathbb{N}$ and maximum degree $k \in [n-1]$, the length of the $k$-PEER HYPER-HYPERCUBE GRAPH is less than or equal to $\max\{1, 2\log_{k+2}(n)\}$.*

*Proof.* We assume that $n$ is decomposed as $n = n_1 \times \cdots \times n_L$ with minimum $L$ where $n_l \in [k+1]$ for all $l \in [L]$. Without loss of generality, we suppose $n_1 \leq n_2 \leq \cdots \leq n_L$. Then, for any $i \neq j$, it holds that $n_i \times n_j \geq k+2$ because if $n_i \times n_j \leq k+1$ for some $i$ and $j$, this contradicts the assumption that $L$ is minimum.

When $L$ is even, we have

$$n = (n_1 \times n_2) \times \cdots \times (n_{L-1} \times n_L) \geq (k+2)^{\frac{L}{2}}.$$

Then, we get $L \leq 2\log_{k+2}(n)$.

Next, we discuss the case when $L$ is odd. When $L \geq 3$, $n_L \geq \sqrt{k+2}$ holds because $n_{L-2} \times n_{L-1} \geq k+2$. Thus, we get

$$n = (n_1 \times n_2) \times \cdots \times (n_{L-2} \times n_{L-1}) \times n_L \geq (k+2)^{\frac{L-1}{2}} \times n_L \geq (k+2)^{\frac{L}{2}}.$$

Then, we get $L \leq 2\log_{k+2}(n)$ when $L \geq 3$.

Thus, given the case when $L = 1$, the length of the $k$-PEER HYPER-HYPERCUBE GRAPH is less than or equal to $\max\{1, 2\log_{k+2}(n)\}$. $\square$

**Lemma 2** (Length of SIMPLE BASE-$(k+1)$ GRAPH). *For any number of nodes $n \in \mathbb{N}$ and maximum degree $k \in [n-1]$, the length of the SIMPLE BASE-$(k+1)$ GRAPH is less than or equal to $2\log_{k+1}(n) + 2$.*

*Proof.* When all prime factors of $n$ are less than or equal to $k+1$, the SIMPLE BASE-$(k+1)$ GRAPH is equivalent to the $k$-PEER HYPER-HYPERCUBE GRAPH and the statement holds from Lemma 1. In the following, we consider the case when there exists a prime factor of $n$ that is larger than $k+1$. Note that because when $L = 1$ (i.e., $n = a_1 \times (k+1)^{p_1}$), all prime factors of $n$ are less than or equal to $k+1$, we only need to consider the case when $L \geq 2$. We have the following inequality:

$$\begin{aligned}
\log_{k+1}(n) &= \log_{k+1}(a_1(k+1)^{p_1} + \cdots + a_L(k+1)^{p_L}) \\
&\geq p_1 + \log_{k+1}(a_1) \\
&\geq p_1.
\end{aligned}$$

Then, because $|V_1| = a_1 \times (k+1)^{p_1}$, it holds that $m_1 = |\mathcal{H}_k(V_1)| \leq 1 + p_1 \leq \log_{k+1}(n) + 1$. Similarly, it holds that $|\mathcal{H}_k(V_{1,1})| = p_1 \leq \log_{k+1}(n)$ because $|V_{1,1}| = (k+1)^{p_1}$. In Alg. 2, the update rule $b_1 \leftarrow b_1 + 1$ in line 22 is executed for the first time when $m = m_1 + 2$ because $L \geq 2$. Thus, the length of the SIMPLE BASE-$(k+1)$ GRAPH is at most $m_1 + |\mathcal{H}_k(V_{1,1})| + 1 \leq 2\log_{k+1}(n) + 2$. This concludes the statement. $\square$

**Lemma 3** (Length of BASE-$(k+1)$ GRAPH). *For any number of nodes $n \in \mathbb{N}$ and maximum degree $k \in [n-1]$, the length of the BASE-$(k+1)$ GRAPH is less than or equal to $2\log_{k+1}(n) + 2$.*

*Proof.* The statement follows immediately from Lemma 2 and line 12 in Alg. 3. $\square$

# E Convergence Rate of DSGD over Various Topologies

Table 2 lists the convergence rates of DSGD over various topologies. These convergence rates can be immediately obtained from Theorem 2 stated in Koloskova et al. [11] and consensus rate of the topology. As seen from Table 2, the BASE-2 GRAPH enables DSGD to converge faster than the ring and torus and as fast as the exponential graph for any number of nodes, although the maximum degree of the BASE-2 GRAPH is only one. Moreover, for any number of nodes, the BASE-$(k+1)$ GRAPH with $2 \le k < \lceil \log_2(n) \rceil$ enables DSGD to converge faster than the exponential graph, even though the maximum degree of the BASE-$(k+1)$ GRAPH remains to be less than that of the exponential graph.

Table 2: Convergence rates and maximum degrees of DSGD over various topologies.

| Topology | Convergence Rate | Maximum Degree | #Nodes $n$ |
|---|---|---|---|
| Ring [28] | $\mathcal{O}\left( \dfrac{\sigma^2}{n\epsilon^2} + \dfrac{\zeta n^2 + \sigma n}{\epsilon^{3/2}} + \dfrac{n^2}{\epsilon} \right) \cdot LF_0$ | 2 | $\forall n \in \mathbb{N}$ |
| Torus [28] | $\mathcal{O}\left( \dfrac{\sigma^2}{n\epsilon^2} + \dfrac{\zeta n + \sigma\sqrt{n}}{\epsilon^{3/2}} + \dfrac{n}{\epsilon} \right) \cdot LF_0$ | 4 | $\forall n \in \mathbb{N}$ |
| Exp. [43] | $\mathcal{O}\left( \dfrac{\sigma^2}{n\epsilon^2} + \dfrac{\zeta \log_2(n) + \sigma\sqrt{\log_2(n)}}{\epsilon^{3/2}} + \dfrac{\log_2(n)}{\epsilon} \right) \cdot LF_0$ | $\lceil \log_2(n) \rceil$ | $\forall n \in \mathbb{N}$ |
| 1-peer Exp. [43] | $\mathcal{O}\left( \dfrac{\sigma^2}{n\epsilon^2} + \dfrac{\zeta \log_2(n) + \sigma\sqrt{\log_2(n)}}{\epsilon^{3/2}} + \dfrac{\log_2(n)}{\epsilon} \right) \cdot LF_0$ | 1 | A power of 2 |
| 1-peer Hypercube [31] | $\mathcal{O}\left( \dfrac{\sigma^2}{n\epsilon^2} + \dfrac{\zeta \log_2(n) + \sigma\sqrt{\log_2(n)}}{\epsilon^{3/2}} + \dfrac{\log_2(n)}{\epsilon} \right) \cdot LF_0$ | 1 | A power of 2 |
| **Base-$(k+1)$ Graph (ours)** | $\mathcal{O}\left( \dfrac{\sigma^2}{n\epsilon^2} + \dfrac{\zeta \log_{k+1}(n) + \sigma\sqrt{\log_{k+1}(n)}}{\epsilon^{3/2}} + \dfrac{\log_{k+1}(n)}{\epsilon} \right) \cdot LF_0$ | $k$ | $\forall n \in \mathbb{N}$ |

# F Additional Experiments

## F.1 Comparison of Base-$(k+1)$ and Simple Base-$(k+1)$ Graphs

Fig. 20 shows the length of the SIMPLE BASE-$(k+1)$ GRAPH and BASE-$(k+1)$ GRAPH. The results indicate that for all $k$, the length of the BASE-$(k+1)$ GRAPH is less than the length of the SIMPLE BASE-$(k+1)$ GRAPH in many cases.

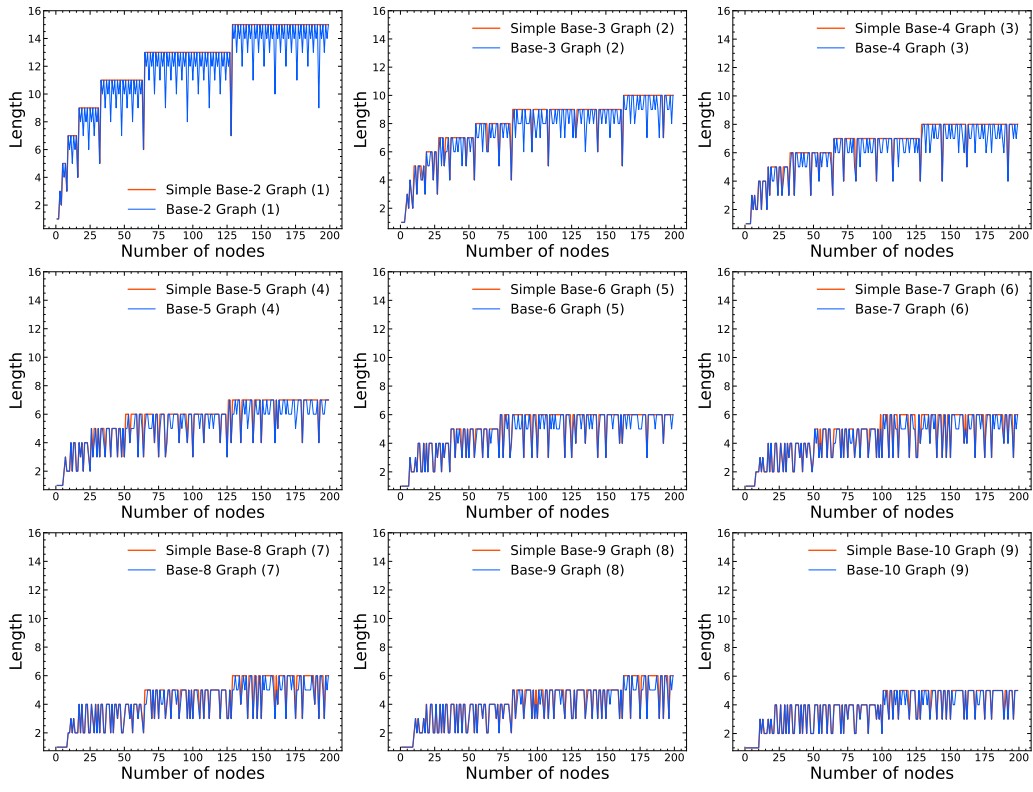

Figure 20: Comparison of the length of the SIMPLE BASE-$(k+1)$ GRAPH and BASE-$(k+1)$ GRAPH.

## F.2 Consensus Rate

In Fig. 21, we demonstrate how consensus error decreases on various topologies when the number of nodes $n$ is a power of 2. The results indicate that the BASE-2 GRAPH and 1-peer exponential graph can reach the exact consensus after the same finite number of iterations and reach the consensus faster than other topologies. Note that the BASE-2 GRAPH is equivalent to the 1-peer hypercube graph when $n$ is a power of 2.

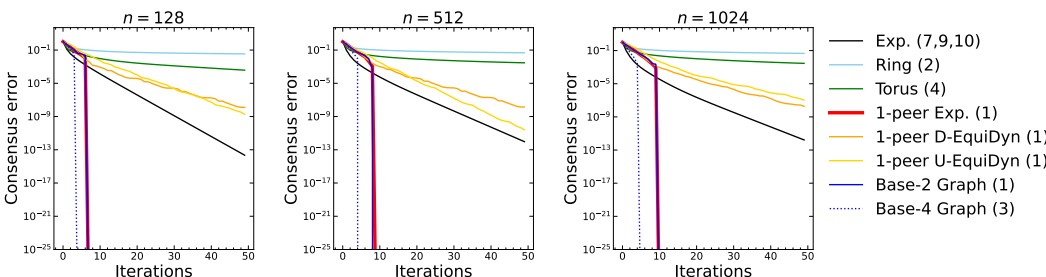

Figure 21: Comparison of consensus rates among different topologies when the number of nodes $n$ is a power of 2. Because the BASE-$\{3,5\}$ GRAPH are the same as the BASE-$\{2,4\}$ GRAPH, respectively, when $n$ is a power of 2, we omit the results of the BASE-$\{3,5\}$ GRAPH.

### F.3 Decentralized Learning

#### F.3.1 Comparison of Base-$(k+1)$ Graph and EquiStatic

In this section, we compared the BASE-$(k+1)$ GRAPH with the {U, D}-EquiStatic [33]. The {U, D}-EquiStatic are dense variants of the 1-peer {U, D}-EquiDyn, and their maximum degree can be set as hyperparameters. We evaluated the {U, D}-EquiStatic varying their maximum degrees; the results are presented in Fig. 22. In both cases with $\alpha = 10$ and $\alpha = 0.1$, the BASE-2 GRAPH can achieve comparable or higher final accuracy than all {U, D}-EquiStatic, and the BASE-$\{3, 4, 5\}$ GRAPH outperforms all {U, D}-EquiStatic. Thus, the BASE-$(k+1)$ GRAPH is superior to the {U, D}-EquiStatic from the perspective of achieving a balance between accuracy and communication efficiency.

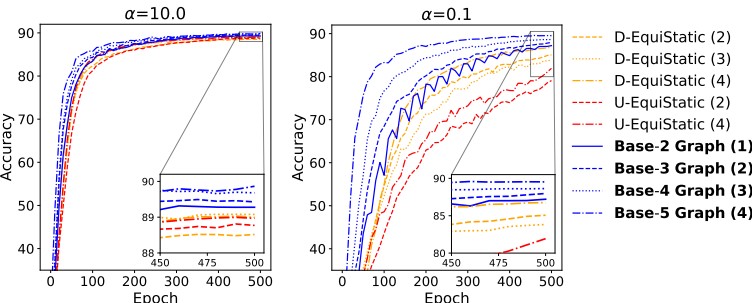

Figure 22: Test accuracy (%) of DSGD with CIFAR-10 and $n = 25$. The number in the bracket is the maximum degree of a topology.

#### F.3.2 Comparison with Various Number of Nodes

In this section, we evaluated the effectiveness of the BASE-$(k+1)$ GRAPH when varying the number of nodes $n$. Fig. 24 presents the learning curves, and Fig 23 shows how consensus error decreases when $n$ is 21, 22, 23, 24, and 25. From Fig. 24, the BASE-2 GRAPH consistently outperforms the 1-peer exponential graph and can achieve a final accuracy comparable to that of the exponential graph. Furthermore, the BASE-$\{3, 4, 5\}$ GRAPH can consistently outperform the exponential graph, even though the maximum degree of the BASE-$\{3, 4, 5\}$ GRAPH is less than that of the exponential graph.

In Fig. 25 presents the learning curve for $n = 16$. When the number of nodes is a power of two, the 1-peer exponential graph is also finite-time convergence, and the 1-peer exponential graph and BASE-2 GRAPH achieve competitive accuracy.

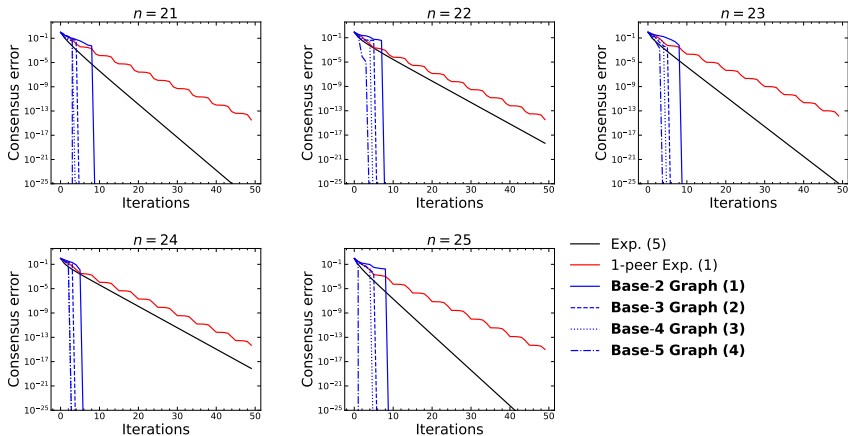

Figure 23: Comparison of consensus rates among different topologies. The number in the bracket denotes the maximum degree of a topology. We omit the results of the BASE-5 GRAPH when $n = 24$ because the BASE-5 GRAPH and BASE-4 GRAPH are equivalent when $n = 24$.

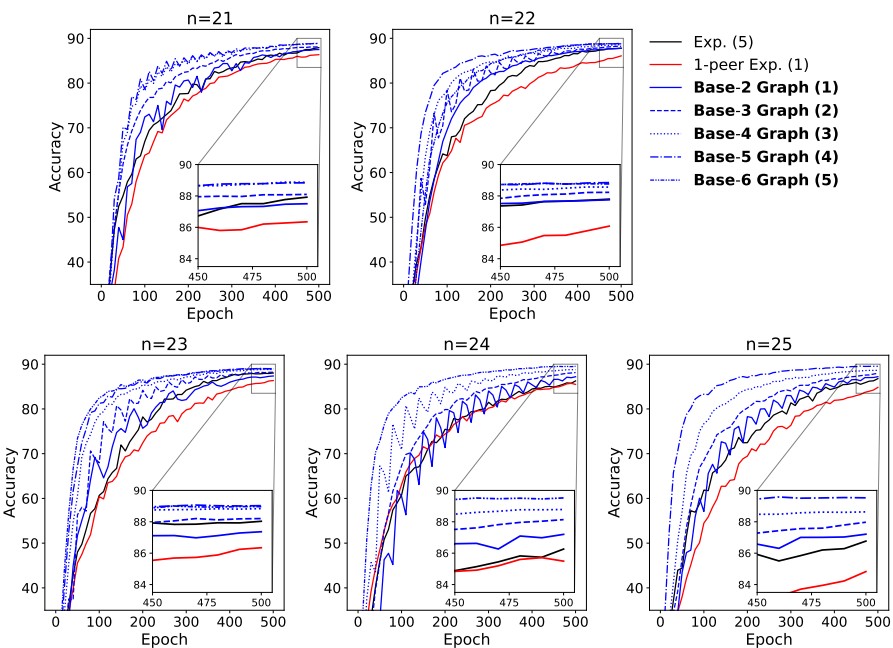

Figure 24: Test accuracy (%) of DSGD with CIFAR-10 and $\alpha = 0.1$. The number in the bracket denotes the maximum degree of a topology. When $n = 24$, we omit the results of the BASE-5 GRAPH because the BASE-5 GRAPH and BASE-4 GRAPH are equivalent. When $n = 25$, we omit the results of the BASE-6 GRAPH because the BASE-6 GRAPH and BASE-5 GRAPH are equivalent.

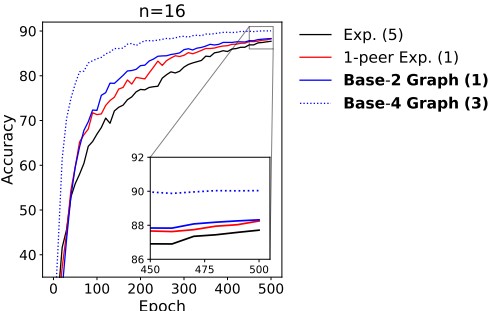

Figure 25: Test accuracy (%) of DSGD with CIFAR-10 and $n = 16$. The number in the bracket is the maximum degree of a topology. We omit the results of the BASE-3 GRAPH and BASE-5 GRAPH because these graphs are equivalent to the BASE-2 GRAPH and BASE-4 GRAPH, respectively.

# G  Results with Other Neural Network Architecture

In this section, we evaluate the effectiveness of the BASE-$(k+1)$ GRAPH with other neural network architecture. Fig. 26 shows the accuracy when we use ResNet-18 [5]. The results are consistent with the results with VGG-11 shown in Sec. 6.

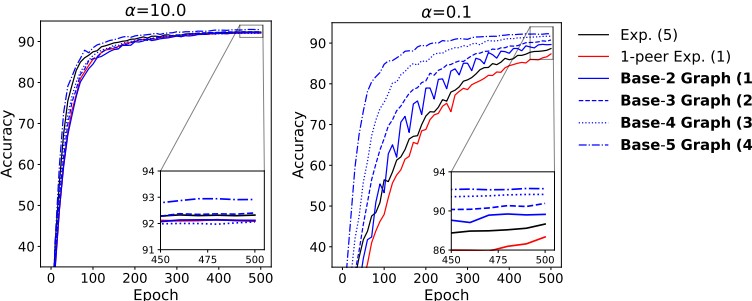

Figure 26: Test accuracy (%) of DSGD with $n = 25$, CIFAR-10, and ResNet. The number in the bracket denotes the maximum degree of a topology.

# H  Hyperparameter Setting

Tables 3 and 4 list the detailed hyperparameter settings used in Secs. 6 and F.3. We ran all experiments on a server with eight Nvidia RTX 3090 GPUs.

Table 3: Hyperparameter settings for Fashion MNIST with LeNet.

| Dataset | Fashion MNIST |
| --- | --- |
| Neural network architecture | LeNet [15] with group normalization [40] |
| Data augmentation | **RandomCrop** of PyTorch |
| Step size | Grid search over $\{0.1, 0.01, 0.001\}$. |
| Momentum | 0.9 |
| Batch size | 32 |
| Step size scheduler | Cosine decay |
| Step size warmup | 10 epochs |
| The number of epochs | 200 |

Table 4: Hyperparameter settings for CIFAR-$\{10, 100\}$ with $\{$VGG-11, ResNet-18$\}$.

| Dataset | CIFAR-$\{10, 100\}$ |
| --- | --- |
| Neural network architecture | $\{$VGG-11 [32], ResNet-18 [5]$\}$ with group normalization [40] |
| Data augmentation | **RandomCrop**, **RandomHorizontalFlip**, **RandomErasing** of PyTorch |
| Step size | Grid search over $\{0.1, 0.01, 0.001\}$. |
| Momentum | 0.9 |
| Batch size | 32 |
| Step size scheduler | Cosine decay |
| Step size warmup | 10 epochs |
| The number of epochs | 500 |

