# OpenReview forum: "Beyond Exponential Graph: Communication-Efficient Topologies for Decentralized Learning via Finite-time Convergence"
_NeurIPS.cc/2023/Conference — NeurIPS 2023 poster_

### Official Review · Reviewer_UfzT · 2023-06-25

**Soundness:** 3 good
**Presentation:** 4 excellent
**Contribution:** 3 good
**Rating:** 6
**Confidence:** 3

**Summary:**

This paper studies communication-efficient topologies for decentralized learning. The proposed topology has four merits: (1) finite-time exact consensus; (2) fast consensus rate (having exact consensus using $O(\log_{k+1} n)$ graphs); (3) arbitrary maximum degree ($1 \leq k \leq n-1$); (4) arbitrary number of nodes. The most related works to this paper are 1-peer exponential graphs, 1-peer hypercube graphs.  Given the existing algorithms, to the best of my knowledge, this paper is the first to get exact consensus using graphs of length $O(\log_{k+1} n)$ instead of $O(\log_2 n)$ with arbitrary number of nodes, where $k$ is the maximum degree. The main algorithm relies on k-peer hyper-hypercube graph. and simple base-(k+1) graph. The k-peer hyper-hypercube graph is a generalization of the 1-peer hypercube graph.  The simple base-(k+1) graph is constructed by dividing the graph into subgraphs with node numbers being a power of k, and then applying the 1-peer hypercube graphs, and using the communication between subgraphs.  The base-(k+1) graphs are constructed by decomposing $n = p\times q$, where $p$ is prime to $q$ and the factors of $q$ are larger than k+1, and then, apply the sub-algorithms respectively.

The paper is generally well-written. The authors provide sufficient illustrations to show how the algorithms work and some numerical experiments to demonstrate the efficiency of their topologies.

**Strengths:**

* the length of graphs needed to have exact consensus is $O(\log_{k+1} n)$, which improves upon the length $O(\log_2 n)$ of the 1-peer exponential graphs.

* the authors use lots of illustrations to show the implementation of the algorithms, which are very clear.

**Weaknesses:**

* The proof for the exact consensus of Algorithm 1 (k-peer hyper hypercube graph) is lacking.

* Considering that the algorithm often divides the graph into many subgraphs, while the nodes do not have equal status, the topologies proposed in this paper are relatively complicated. The intricate graph structures could possibly limit its applications in practice, like delayed communication, compressed communication, and information loss.

* The descriptions of the algorithm are only from a panoramic view. It is not clear whether there is a simple way for each node to know its neighbor in each graph efficiently.

**Questions:**

* Is it possible to give theoretical guarantees for the claim that base-(k+1) graphs have shorter lengths than simple base-(k+1) graphs?

* Are there efficient methods for each node to know its neighbors in each graph instead of generating the entire graph by each node?

* Is it possible to extend the algorithm to more restricted topologies where each node can only communicate with a subset of nodes directly?

**Limitations:**

I would suggest the authors provide formal proof, or at least a sketch of proof, for the exact consensus of algorithm 1 (k-peer hyper hypercube graph).

---

> ### Author Rebuttal · Authors · 2023-08-09
>
>
> > The proof for the exact consensus of Algorithm 1 (k-peer hyper hypercube graph) is lacking.
>
> It is explained that the $k$-peer hyper-hypercube graph is the finite-time convergence in Sec. 4.1.
> In the camera-ready version, we will add rigorous proof to show that the $k$-peer hyper-hypercube graph is finite-time convergence.
>
>
> > Is it possible to give theoretical guarantees for the claim that base-(k+1) graphs have shorter lengths than simple base-(k+1) graphs?
>
> Thank you for pointing this out.
> Thanks to line 12 in Alg. 3, the length of the Base-($k+1$) Graph is guaranteed to be less than or equal to that of the Simple Base-($k+1$) Graph.
> To guarantee that the Base-$(k+1)$ Graph has the shorter or same length as the Simple Base-($k+1$) Graph,
> line 12 is necessary.
> If we omit line 12, there exists the case where the Base-$(k+1)$ Graph is longer than the Simple Base-$(k+1)$ Graph, e.g., $(n, k) = (22, 4)$.
>
> > Are there efficient methods for each node to know its neighbors in each graph instead of generating the entire graph by each node?
>
> It is possible to rewrite Algs. 2 and 3 as in Alg. 1,
> and if we rewrite Algs. 2 and 3 as in Alg. 1, the neighbors' indexes can be written as in line 7 in Alg. 1.
>
> > Is it possible to extend the algorithm to more restricted topologies where each node can only communicate with a subset of nodes directly?
>
> Extending the Base-$(k+1)$ Graph to the setting the reviewer mentioned is not trivial.
> However, we believe that the key idea of the Base-$(k+1)$ Graph can be used to construct such restricted topologies
> because the idea of splitting the set of nodes into subsets where the $k$-peer hyper-hypercube graph is applicable is a simple and efficient method.

---

> > ### Comment · Reviewer_UfzT · 2023-08-20
> > **Thanks for the response.**
> >
> > Thanks for the clarification.

---

### Official Review · Reviewer_hhaz · 2023-07-04

**Soundness:** 3 good
**Presentation:** 4 excellent
**Contribution:** 3 good
**Rating:** 6
**Confidence:** 3

**Summary:**

This paper introduces time-varying communication topologies to tackle the problem of communication efficiency in decentralized learning. These topologies, inspired by the idea of the 1-peer exponential graph, achieve finite-time exact consensus, meaning that the exact average of values is obtained after a finite number of steps.

The graph is obtained through an iterative process: starting from the k-peer hyper-hypercube graph (an extension of the 1-peer hypercube graph), the simple-(k+1) graph is introduced to handle the case when $n$ has large prime factors, using k-peer hyper-hypercube graph as a subroutine. Finally, the base-(k+1) algorithm is developed based on the two previous graphs to remove redundancy and obtain exact consensus in fewer iterations.

Then, a convergence result is given based on the bound from Koloskova et al. (2020b), and experimental results suggest that the base-(k+1) graph improves over the (1-peer) exponential graph indeed. Unsurprisingly, more connected topologies lead to better accuracies in heterogeneous settings.

**Strengths:**

- Finite-time consensus for any number of nodes.
- The bandwidth / speed trade-off can be controlled through parameter k (max degree at each iteration).
- Great efforts are made to make the constructions understandable (though they still remain quite hard to grasp), through figures and step-by-step introduction.

**Weaknesses:**

- I find the state of the art is not discussed in enough details. While decentralized methods are discussed in details, the finite-time consensus aspect calls to comparing with efficient allreduce methods as well, that are used to scale modern deep learning algorithms. I am not an expert in this domain, but a quick search led me to, e.g., refs [A,B,C] (but there are probably many relevant others).

The advantage of finite-time consensus is that updates can be done after each iteration, which is not necessarily the case with allreduce methods. At the same time, there are no guarantee on how close to the mean the iterates are before finite-time convergence, so several SGD steps without communication could also be performed between allreduce SGD for instance (local SGD style), which would not be too different. This could / should also be compared against.

Minor comment: One can always have a number of nodes equal to a power of 2 by introducing some nodes with value 0 (and upweighing the other nodes), and making it so that each actual node owns at most 1 virtual node, thus at most doubling the effective degree. It's an easy way to use 1-peer exponential graphs for all n.

[A] Rabenseifner, Rolf. "Optimization of collective reduction operations." Computational Science-ICCS 2004: 4th International Conference, Kraków, Poland, June 6-9, 2004, Proceedings, Part I 4. Springer Berlin Heidelberg, 2004.

[B] Mikami, Hiroaki, et al. "Massively distributed SGD: ImageNet/ResNet-50 training in a flash." arXiv preprint arXiv:1811.05233 (2018).

[C] Ueno, Yuichiro, and Rio Yokota. "Exhaustive study of hierarchical allreduce patterns for large messages between gpus." 2019 19th IEEE/ACM International Symposium on Cluster, Cloud and Grid Computing (CCGRID). IEEE, 2019.

**Questions:**

- Can you clarify how your method compares with optimized allreduce methods? For instance if you use it until exact consensus each time. What are the bandwidth/latency trade-offs achieved? How do they compare to existing results?
- How robust is the Base-(k+1) graph approach to failures/packet drops? Finite-time exact consensus would be lost in this case, but can we recover good approximation properties or is everything lost in this case?

---

> ### Author Rebuttal · Authors · 2023-08-09
>
> > [...] While decentralized methods are discussed in details, the finite-time consensus aspect calls to
> comparing with efficient allreduce methods as well, [...] The advantage of finite-time consensus is that updates can be done after each iteration, which is not necessarily the case with allreduce methods. At the same time, there are no guarantee on how close to the mean the iterates are before finite-time convergence, so several SGD steps without communication could also be performed between allreduce SGD for instance (local SGD style), which would not be too different. This could / should also be compared against.
>
> One potential advantage of the Base-$(k+1)$ Graph over All-Reduce with local steps is that
> when communication failures occur between nodes,
> All-Reduce with local steps fails to converge to the stationary point, whereas the Base-$(k+1)$ Graph remains to converge.
> More specifically, as we mentioned in line 97, most of the decentralized learning methods require the mixing matrix $\mathbf{W}$ to be doubly stochastic (i.e., $\mathbf{W} \mathbf{1}_n = \mathbf{1}_n$ and $\mathbf{W}^\top \mathbf{1}_n = \mathbf{1}_n$,
> and the average parameters that nodes have are the same before and after the communication.).
> Intermediate steps of All-Reduce do not entail doubly-stochastic mixing matrices, though the entire sequence of All-Reduce communication does.
> Hence, even a single communication failure prevents us from guaranteeing convergence because the average parameters change before and after the communication.
> On the other hand, even if communication failures occur, the Base-$(k+1)$ Graph satisfies this condition
> and enable decentralized learning methods to converge.
>
> Although it is beyond the scope of our paper to propose topologies that are robust to communication failures,
> our work will be an important step in the research to create topologies with high communication efficiency, fast convergence rate, and robustness to communication failures.
>
> > One can always have a number of nodes equal to a power of 2 by introducing some nodes with value 0 (and upweighing the other nodes), and
> making it so that each actual node owns at most 1 virtual node, thus at most doubling the effective degree. It's an easy way to use 1-peer exponential graphs for all
> n.
>
> Thank you for the comments.
> The reviewer's idea is also finite-time convergence for any number of nodes,
> but servers need to have two parameters, the parameters of the actual and virtual nodes.
> Thus, the Base-$(k+1)$ Graph is more memory efficient than the reviewer's idea.
> Additionally, the $1$-peer exponential graph is finite-time convergence only when the maximum degree is one,
> whereas the Base-$(k+1)$ Graph is finite-time convergence for any maximum degree.
> It is also an important advantage of the Base-($k+1$) Graph over the reviewer's idea.
>
> > Can you clarify how your method compares with optimized allreduce methods? For instance if you use it until exact consensus each time. What are the
> bandwidth/latency trade-offs achieved? How do they compare to existing results?
>
> Please see the first response.
>
> > How robust is the Base-(k+1) graph approach to failures/packet drops? Finite-time exact consensus would be lost in this case, but can we recover good
> approximation properties or is everything lost in this case?
>
> Even if communication failures occur,
> the Base-$(k+1)$ Graph makes Decentralized SGD converge to the stationary point if there exists a path between any two nodes.
> However, if communication failures occur, the Base-($k+1$) Graph is no longer finite-time convergence
> and cannot make Decentralized SGD converge with the rate shown in Theorem 2.
> It is an interesting future work to propose the topologies that are finite-time convergence even if communication failures occur,
> but it is beyond the scope of our paper.

---

### Official Review · Reviewer_BhdZ · 2023-07-07

**Soundness:** 3 good
**Presentation:** 3 good
**Contribution:** 3 good
**Rating:** 5
**Confidence:** 5

**Summary:**

This paper introduces the BASE-(k+1) GRAPH, a new way of designing topology for decentralized learning that is shown to enjoy a fast consensus rate while maintaining small maximum degree. The authors validate the effectiveness of the BASE-(k+1) GRAPH through experiments, demonstrating its finite-time convergence for certain number of nodes and its effectiveness for decentralized learning. Overall, the BASE-(k + 1) GRAPH offers improved communication efficiency and convergence rate for decentralized learning methods.

**Strengths:**

- The paper proposes a new way of designing topology for decentralized learning, the BASE-(k + 1) GRAPH, which allows for a fast consensus rate while maintaining small maximum degree to enable more communication-efficient decentralized learning.

- The authors provide a theoretical analysis of the proposed method which ensures the property of small maximum degree and several experiments are conducted to validate their claims.

- The paper is well written and organized, with a logical flow of ideas. The authors provide a clear explanation of the proposed approach, and also includes helpful figures and tables to illustrate the concepts and results.



**Weaknesses:**

- The numerical experiments are still not sufficient; it would be more convincing if they compared the BASE-(k + 1) GRAPH with a more comprehensive set of topologies to demonstrate its superiority. As far as the reviewer know, there are several other topologies that can also achieve fast consensus rate and small maximum degree, such as Ring All-Reduce where maximum degree is 1 [a] , Tree All-Reduce where maximum degree is 2 [b] which and optimization-based topology [c].

- The paper does not explain how to decompose n, whether different decomposition methods will produce different results and how to decompose n to get the best results.

- The paper assumes a homogeneous network, which may not be practical in some real-world scenarios, where communication bandwidth and underlying topology constraints may affect the communication between nodes.

[a] Alexander Sergeev and Mike Del Balso. Horovod: fast and easy distributed deep learning in tensorflow. arXiv preprint arXiv:1802.05799, 2018.

[b] https://developer.nvidia.com/blog/massively-scale-deep-learning-training-nccl-2-4/

[c] Sun, Chuangchuang, Ran Dai, and Mehran Mesbahi. "Weighted network design with cardinality constraints via alternating direction method of multipliers." IEEE Transactions on Control of Network Systems 5.4 (2018): 2073-2084.


**Questions:**

- The reviewer would also like to know how the method can be extended to heterogeneous networks where the topology may not exhibit regular structures.

- There are several other topologies that can also achieve fast consensus rate and small maximum degree; how the proposed method compares with them?

- For a given number of nodes n, there exists several ways of decomposition, and different way of decomposition will lead to different maximum degree and consensus rate. The reviewer would like to know how one can find the optimal decomposition method to properly balance the maximum degree and consensus rate.


**Limitations:**

- The paper could benefit from a more detailed discussion of the limitations of the proposed approach, such as its sensitivity to certain types of data heterogeneity or its applicability to heterogeneous networks.

- A more detailed description of the algorithm and its implementation (such as the decomposition method of node number n) would be helpful for readers who want to replicate the experiments or apply the method to their own problems.

- The BASE-(k + 1) GRAPH should be compared with a wider range of existing topologies.

- The derivation of Theorem 2 should be shown to help readers better understand the influence of BASE-(k + 1) GRAPH on the convergence of decentralized SGD.

---

> ### Author Rebuttal · Authors · 2023-08-09
>
> > The numerical experiments are still not sufficient; it would be more convincing if they compared the BASE-(k + 1) GRAPH with a more comprehensive set of
> topologies to demonstrate its superiority. As far as the reviewer know, there are several other topologies that can also achieve fast consensus rate and smal
> maximum degree, such as Ring All-Reduce where maximum degree is 1 [a] , Tree All-Reduce where maximum degree is 2 [b] which and optimization-based
> topology [c].
>
> Ring All-Reduce and Tree All-Reduce can also be regarded as the finite-time convergent sequence of graphs whose maximum degree is one and two, respectively.
> Specifically, Ring All-Reduce can be regarded as the sequence of $n$ graphs,
> where $n$ is the number of nodes.
> However, as mentioned in line 97, most of the decentralized learning methods require the mixing matrix $\mathbf{W}$ to be doubly stochastic (i.e., $\mathbf{W} \mathbf{1}_n = \mathbf{1}_n$ and $\mathbf{W}^\top \mathbf{1}_n = \mathbf{1}_n$,
> and the average parameters that nodes have are the same before and after the communication).
> Then, the entire sequence of Ring All-Reduce satisfies this condition, but each graph in Ring All-Reduce does not satisfy this condition.
> Therefore, the sequence of graphs associated with Ring All-Reduce and Tree All-Reduce can not be used as the underlying network topology of decentralized learning methods.
>
> > The paper does not explain how to decompose n, whether different decomposition methods will produce different results and how to decompose n to get the
> best results.
>
> The main proposed method is the Base-($k+1$) Graph,
> and Sec. 4 and Algs. 1, 2, and 3 explain how to construct the Base-($k+1$) Graph.
> The $k$-peer hyper-hypercube and Simple Base-$(k+1)$ Graphs are topologies introduced to illustrate the key ideas of the Base-$(k+1)$ Graph.
> Then, in Sec. B.3, we compare the Base-($k+1$) Graph with the Simple Base-($k+1$) Graph by showing many illustrations.
> In Figs. 5 and 19, we demonstrate that the length of the Base-($k+1$) Graph is less than that of the Simple Base-($k+1$) Graph.
>
> > The paper assumes a homogeneous network, which may not be practical in some real-world scenarios, where communication bandwidth and underlying
> topology constraints may affect the communication between nodes.
>
> The prior works showed that communication efficiency is determined by the maximum degree of underlying network topologies in the homogeneous network setting (Neglia et al., 2019; Ying et al., 2021; Wang et al., 2019).
> Following these prior works,
> our work proposed novel topologies that can more successfully reconcile the convergence rates and communication efficiency than the existing topologies.
> It is beyond the scope of our paper to address the heterogeneous network,
> but the key idea of the Base-($k+1$) Graph may also be used to address the heterogeneous network setting
> because our key idea of splitting the set of nodes into subsets is a simple and fundamental method.
>
> > The reviewer would also like to know how the method can be extended to heterogeneous networks where the topology may not exhibit regular structures.
>
> The Base-$(k+1)$ Graph is constructed with the constraint that its maximum degree is less than or equal to $k$.
> To address the heterogeneous bandwidth network,
> it is interesting future research to extend the Base-($k+1$) Graph such that the degree of each node is less than the given hyperparameter, which can differ for each node.
> It is not trivial to extend the Base-$(k+1)$ Graph to such heterogeneous network settings.
> However, the key idea of the Base-$(k+1)$ Graph may also be applicable
> because our key idea of splitting the set of nodes into subsets is a simple and efficient method.
>
>
> > There are several other topologies that can also achieve a fast consensus rate and small maximum degree; how the proposed method compares with them?
>
> Please see the first response.
>
> > For a given number of nodes n, there exists several ways of decomposition, [...] The reviewer would like to know how one can find the optimal decomposition method to properly balance the maximum degree and consensus
> rate.
>
> Please see the second response.
>
> > A more detailed description of the algorithm and its implementation (such as the decomposition method of node number n) would be helpful for readers who want to replicate the experiments or apply the method to their own problems.
>
> We will make our implementation public and include the GitHub repository URL in the camera-ready version so that readers can reproduce our proposed method.
>
> > The derivation of Theorem 2 should be shown to help readers better understand the influence of BASE-(k + 1) GRAPH on the convergence of decentralized SGD.
>
> Let $\mathbf{W}^{(1)}, \cdots, \mathbf{W}^{(m)}$ be mixing matrices of the Base-($k+1$) Graph.
> These mixing matrices satisfy the following for any $\mathbf{X} \in \mathbb{R}^{d \times n}$:
>
> $$
>     \| \mathbf{X} \prod_{k=1}^m \mathbf{W}^{(k)} - \bar{\mathbf{X}} \|^2_F = 0,
> $$
> where $\bar{\mathbf{X}} = \frac{1}{n} \mathbf{X} \mathbf{1}_n \mathbf{1}_n^\top$.
> Koloskova et al. (2020) derived the convergence rate of Decentralized SGD with time-varying topologies.
> Thus, substituting the above equation into Theorem 2 in (Koloskova et al. (2020)) yields our Theorem 2.
>
> ### Reference
> Koloskova et al., (2020). A unified theory of decentralized SGD with changing topology and local updates. In International Conference on Machine Learning
>
> Neglia et al., (2019). The role of network topology for distributed machine learning. In IEEE Conference on Computer Communications.
>
> Wang et al., (2019). Matcha: Speeding up decentralized sgd via matching decomposition sampling. In Indian Control Conference.
>
> Ying et al., (2021). Exponential graph is provably efficient for decentralized deep training. In Advances in Neural Information Processing Systems

---

> > ### Comment · Reviewer_BhdZ · 2023-08-13
> >
> > The reviewer would like to thank the authors' response which have addressed most of the concerns. However, the reviewer disagrees with that "...the sequence of graphs associated with Ring All-Reduce and Tree All-Reduce can not be used as the underlying network topology of decentralized learning methods" as these topologies are widely used in dencentralized learning (both in academia and industry). Thus, the reviewer still believe that it is important to compare the experimental results with that of Ring All-Reduce/Tree All-Reduce to show its advantages. In this regards, the reviewer would keep the rate unchanged for now.

---

### Official Review · Reviewer_pdDT · 2023-07-14

**Soundness:** 3 good
**Presentation:** 4 excellent
**Contribution:** 4 excellent
**Rating:** 7
**Confidence:** 4

**Summary:**

The paper considers the problem of decentralized optimization and proposes a novel time-varying communication topology that has better properties than prior topologies, especially when the number of nodes n is not a power of 2 i.e. (i) it is sparse, and (ii) has good spectral properties, translating to the good optimization behavior. Paper provides a description of the proposed topology, its properties, and evaluates its effectiveness in practice.


**Strengths:**

- The paper is very well written and clear
- Proposed topology clearly improves over the previously known topologies, especially when the number of nodes n is not a power of 2.


**Weaknesses:**

1. Paper gives consensus rate (in e.g. Table 1) in O-notation, i.e. hiding the constants. However, actual constants do matter a lot. If we allow the use of O-notation, then expander graphs would be the best theoretical graphs, as they have constant consensus rate, as well as constant maximum degree. However, expanders do not work well in practice for relatively small graphs as the constants are quite large. Therefore, I believe it is important to get good constants while designing a new communication topology.

2. Description of the algorithm for Base-(k + 1) graph is not fully clear. In step 1. of the algorithm (line 181), it was not clear why n can always be decomposed in this way? What if n is a prime number ?


**Questions:**

1. If the number of nodes n is a power of two, is it still beneficial to use your proposed topologies ? Including the larger range of n in figure 8 that covers power of 2 might be good.
2. Why did you not include the gradient tracking in comparison in figure 9?
3. When reading Figure 3, I got confused why the number of averaging steps is 5, while log_2(5) < 3.
4. In the Experiments section (Fig. 6, 7, 8, 9), having a Base-6 graph might still potentially be better than the Exponential graph, as its maximum degree would be the same, but it would lead to the faster convergence. Why did you not include the Base-6 graph in the comparison?

Minor suggestions:

5. It might be good to provide the open-source code of generating proposed topologies.
6. Might be helpful for the reader to indicate how large is log_2(n) for each of the experiments.
7. When giving an example in Figures 3 & 4, I felt it might be better to have the same number of nodes n in both cases, so that the reader can see how is the Base-(k + 1) graph can reduce the number of steps from the Simple Base-(k + 1) graph

**Limitations:**

yes

---

> ### Author Rebuttal · Authors · 2023-08-09
>
> > Paper gives consensus rate (in e.g. Table 1) in O-notation, i.e. hiding the constants. However, actual constants do matter a lot. If we allow the use of O-notation,
> then expander graphs would be the best theoretical graphs, as they have constant consensus rate, as well as constant maximum degree. However, expanders do
> not work well in practice for relatively small graphs as the constants are quite large. Therefore, I believe it is important to get good constants while designing a
> new communication topology.
>
> Theorem 1 provides the upper bound of the length of the Base-$(k+1)$ Graph without using O-notation,
> showing that the Base-$(k+1)$ Graph is $(2 \log_{k+1} (n) + 2)$-finite time convergence.
> Thus, the constant factors are small,
> and the experimental results also demonstrate that the Base-$(k+1)$ Graph enables decentralized learning methods to more successfully reconcile accuracy and communication efficiency than the existing topologies.
>
> > Description of the algorithm for Base-(k + 1) graph is not fully clear. In step 1. of the algorithm (line 181), it was not clear why n can always be decomposed in this
> way? What if n is a prime number ?
>
> When the number of nodes $n$ is a prime number, $(p, q)$ in line 2 in Alg. 3 becomes $(1, n)$ or $(n, 1)$,
> and then the Base-$(k+1)$ Graph becomes equivalent to the Simple Base-($k+1$) Graph.
> For instance, when $(p, q)=(1,n)$, $\mathcal{E}$ in line 11 is the graph sequence consisting of the Simple Base-$(k+1)$ Graph $A^{\text{simple}}_k (V)$ and an empty graph.
> Then, thanks to the condition in line 12, the Base-($k+1$) Graph becomes equivalent to the Simple Base-($k+1$) Graph.
> Sec. B.3 shows examples when the number of nodes is a prime number.
>
> > If the number of nodes n is a power of two, is it still beneficial to use your proposed topologies ? Including the larger range of n in figure 8 that covers power of 2
> might be good.
>
> The $1$-peer hypercube and $1$-peer exponential graphs are finite-time convergence only when the maximum degree is one,
> whereas the Base-($k+1$) Graph is finite-time convergence for any maximum degree $k$.
> Thus, if we need to use the topology with a fast consensus rate due to the high data heterogeneity,
> it is still beneficial to use the Base-$(k+1)$ Graph even when the number of nodes is a power of two.
>
> In the camera-ready version, we will add the experimental results to evaluate the Base-$(k+1)$, $1$-peer hypercube, and $1$-peer exponential graphs in the setting where the number of nodes is a power of two.
>
> > Why did you not include the gradient tracking in comparison in figure 9?
>
> Communication costs are one of the most important factors in decentralized learning,
> and Gradient Tracking requires twice as large communication costs as Decentralized SGD, $D^2$, and QG-DSGDm.
> Thus, we evaluated our proposed methods with Decentralized SGD, $D^2$, and QG-DSGDm instead of Gradient Tracking.
>
> > When reading Figure 3, I got confused why the number of averaging steps is 5, while $\log_2(5) < 3$.
>
> As shown in Theorem 1, the upper bound of the length of the Simple Base-($k+1$) and Base-($k+1$) Graphs are not $\log_{k+1} (n)$, but $2 \log_{k+1} (n) + 2$.
>
> > In the Experiments section (Fig. 6, 7, 8, 9), having a Base-6 graph might still potentially be better than the Exponential graph, as its maximum degree would be
> the same, but it would lead to the faster convergence. Why did you not include the Base-6 graph in the comparison?
>
>
> We evaluated the Base-$6$ Graph with CIFAR-10, $\alpha=0.1$, and $n=24$,
> showing the results in the following table.
> Note that when $n=25$, the Base-$6$ Graph becomes equivalent to the Base-$5$ Graph.
> The results indicate that the Base-$6$ Graph can further improve the accuracy.
> We will evaluate the Base-$6$ Graph with other $n$ and add the experimental results in the camera-ready version.
>
> | Topology     | Accuracy        |
> |--------------|-----------------|
> | Exp.         | $86.3 \pm 1.14$ |
> | 1-peer Exp.  | $85.5 \pm 1.10$ |
> | Base-2 Graph | $87.2 \pm 0.53$ |
> | Base-3 Graph | $88.1 \pm 0.23$ |
> | Base-4 Graph | $88.8 \pm 0.07$ |
> | Base-5 Graph | $88.8 \pm 0.07$ |
> | Base-6 Graph | $89.7 \pm 0.34$ |
>
>
> > It might be good to provide the open-source code of generating proposed topologies.
>
> Our code is contained in the supplementary material. We promise to make it public and add the GitHub repository URL in the camera-ready version.
>
>
> > When giving an example in Figures 3 \& 4, I felt it might be better to have the same number of nodes n in both cases, so that the reader can see how is the Base-
> (k + 1) graph can reduce the number of steps from the Simple Base-(k + 1) graph.
>
> We will add the illustration of the Simple Base-$2$ Graph with $n=6$ in Figs. $3$ or $4$ in the camera-ready version.

---

> > ### Comment · Reviewer_pdDT · 2023-08-16
> >
> > I would like to thank the authors for their explanations. I do not have further questions.

---

### Decision · Program_Chairs · 2023-09-21

**Decision:**

Accept (poster)

**Comment:**

This paper delves into a significant inquiry concerning the design of communication graphs, with a focus on the "Base-k" graph, which facilitates swift consensus convergence through a finite number of gossip iterations. All reviewers acknowledge the paper's clarity and results. I'd like to emphasize the value of the comparison suggested by reviewer BhdZ, involving standard communication topology, and recommends its incorporation into the final version of this paper.